# Effects of therapeutic vaccination on the control of SIV in rhesus macaques with variable responsiveness to antiretroviral drugs

Hillary Claire Tunggal[1,2], Paul Veness Munson[1,2¤a], Megan Ashley O'Connor[1,2], Nika Hajari[2], Sandra Elizabeth Dross[1,2], Debra Bratt[2¤b], James Thomas Fuller[1], Kenneth Bagley[3¤c], Deborah Heydenburg Fuller[1,2]*

1 Department of Microbiology, University of Washington, Seattle, Washington, United States of America,
2 Washington National Primate Research Center, Seattle, Washington, United States of America,
3 Profectus Biosciences, Baltimore, Maryland, United States of America

¤a Current address: Parker Institute for Cancer Immunotherapy, San Francisco, California, United States of America
¤b Current address: University of Georgia, Athens, Georgia, United States of America
¤c Current address: Orlance Incorporation, Baltimore, Maryland, United States of America
* fullerdh@uw.edu

**Data Availability Statement:** All relevant data are within the manuscript and its Supporting information files.

## Abstract

A therapeutic vaccine that induces lasting control of HIV infection could eliminate the need for lifelong adherence to antiretroviral therapy. This study investigated a therapeutic DNA vaccine delivered with a single adjuvant or a novel combination of adjuvants to augment T cell immunity in the blood and gut-associated lymphoid tissue in SIV-infected rhesus macaques. Animals that received DNA vaccines expressing SIV proteins, combined with plasmids expressing adjuvants designed to increase peripheral and mucosal T cell responses, including the catalytic subunit of the *E. coli* heat-labile enterotoxin, IL-12, IL-33, retinaldehyde dehydrogenase 2, soluble PD-1 and soluble CD80, were compared to mock-vaccinated controls. Following treatment interruption, macaques exhibited variable levels of viral rebound, with four animals from the vaccinated groups and one animal from the control group controlling virus at median levels of $10^3$ RNA copies/ml or lower (controllers) and nine animals, among all groups, exhibiting immediate viral rebound and median viral loads greater than $10^3$ RNA copies/ml (non-controllers). Although there was no significant difference between the vaccinated and control groups in protection from viral rebound, the variable virological outcomes during treatment interruption enabled an examination of immune correlates of viral replication in controllers versus non-controllers regardless of vaccination status. Lower viral burden in controllers correlated with increased polyfunctional SIV-specific CD8[+] T cells in mesenteric lymph nodes and blood prior to and during treatment interruption. Notably, higher frequencies of colonic CD4[+] T cells and lower Th17/Treg ratios prior to infection in controllers correlated with improved responses to ART and control of viral rebound. These results indicate that mucosal immune responses, present prior to infection, can influence efficacy of antiretroviral therapy and the outcome of immunotherapeutic

**Funding:** This work was supported with federal funds from the National Institutes of Health (www. nih.gov, T32-AI007140 to M.A.O.), the National Institute of Allergy and Infectious Diseases (www. niaid.nih.gov, R44AI110315 to K.B. and R01 AI104679 to D.H.F.), and the Office of Research Infrastructure Programs (orip.nih.gov, P51OD010425 to D.H.F.). During this study, K. B. was a paid employee of Profectus Biosciences and assisted in study design and participated in the review and editing of this manuscript. Apart from these specific roles, Profectus Biosciences did not have any additional role in the data collection, data analysis or decision to publish the manuscript. K.B. is currently a paid employee of Orlance Incorporation. Orlance did not provide any support for this study and was not involved in the study design, data collection and analysis, decision to publish, or preparation of the manuscript. The other funders of this study provided support in the form of salaries for authors H.C.T., P.V.M., M.A.O., N.H., S.E.D., J.T.F., D.B., and D.H.F., but did not have any additional role in the study design, data collection and analysis, decision to publish, or preparation of the manuscript. The specific roles of these authors are articulated in the 'author contributions' section.

**Competing interests:** Dr. Kenneth Bagley was a paid employee of Profectus Biosciences and had stock options with Profectus when the research was performed. A portion of this study was supported by the SBIR R44AI110315 awarded to Profectus BioSciences. Dr. Bagley was the Principal Investigator for R44AI110315. At the time the study was performed, Dr. Bagley had unexercised stock options with with Profectus BioSciences. Dr. Bagley left Profectus BioSciences in August 2019. He did not exercise any stock options and no longer has an interest in Profectus BioSciences. Dr. Kenneth Bagley is currently a paid employee of Orlance Incorporated, which was co-founded by D.H.F. This study was not supported in any way by Orlance. The affiliations of Dr. Bagley with Profectus and Orlance and Dr. Fuller's co-founder interests in Orlance do not alter our adherence to PLOS ONE policies on sharing data and materials. There are no patent applications (pending or actual) affiliated with this study.

vaccination, suggesting that therapies capable of modulating host mucosal responses may be needed to achieve HIV cure.

## Introduction

ART greatly reduces HIV replication and restores CD4$^+$ T cell counts, thus preventing progression to AIDS and prolonging the lifespan of people living with HIV [1]. However, ART alone is unable to eliminate the latent viral reservoir, which necessitates strict lifelong adherence to a daily ART regimen [2]. For most individuals, ART interruption will lead to a resurgence in viral replication within weeks [3]. However, continuous usage of ART can be prohibitively expensive and may result in side effects that discourage compliance [4,5]. Furthermore, ART cannot fully reverse the immune dysfunction induced by HIV, particularly in the gut mucosa, that drives chronic immune activation and disease pathogenesis [6,7]. Thus, although advances in ART have greatly improved the health and life expectancy of people living with HIV, a vaccine or cure for HIV is still urgently needed, especially for people in developing countries that are most affected by this pandemic.

To this end, many cure strategies are in development, including therapeutic HIV vaccines designed to enhance virus-specific T-cellular and humoral immune responses to provide immune control of virus replication after stopping ART. Numerous therapeutic HIV vaccines have been tested, both in the SIV/SHIV nonhuman primate (NHP) model and in human clinical trials, including protein subunit [8,9], live-attenuated [10], dendritic cell [11], viral vectored [12,13], and DNA vaccines [14–16]. Unfortunately, none of these approaches have resulted in durable control of viremia in human clinical trials and immunotherapies in NHP have, at best, achieved approximately 50% efficacy [17–19]. This suggests that inherent host factors may impact the efficacy of therapeutic interventions but, to date, there is an incomplete understanding of what these factors are.

The gut is a major site of HIV and SIV replication [20–22], resulting in depletion and functional alteration of gut mucosal CD4$^+$ T cells and loss of antigen-presenting cells and innate lymphocytes [23]. These events contribute to structural damage of the gastrointestinal (GI) tract and systemic translocation of GI microbial products that drives chronic immune activation and disease pathogenesis [24,25]. We previously showed in the rhesus macaque model that an SIV DNA vaccine expressing SIV Gag, RT, Nef and Env and co-delivered with a plasmid expressing a mucosal adjuvant, the heat-labile *E. coli* enterotoxin (LT) by particle-mediated epidermal delivery (PMED or gene gun), induced durable protection from viral rebound and disease progression after ART withdrawal in approximately 60% of animals. In that study, Gag-specific mucosal T cell responses in the vaccinated animals significantly correlated with reduced viremia [15], suggesting that SIV-specific T cell responses in the gut are important for controlling viral rebound. To further improve therapeutic efficacy, we therefore tested a new multiantigen SIV DNA vaccine (MAG) expressing Gag, Pol and Env, delivered by intradermal electroporation with a novel combination of adjuvants designed to increase both mucosal and systemic immunogenicity in SIV-infected rhesus macaques. Our adjuvant combination (AC) consisted of co-delivered DNA plasmids encoding the catalytic subunit of LT (LTA1), the cytokines IL-12 and IL-33, the enzyme retinaldehyde dehydrogenase 2 (RALDH2), soluble PD-1 (sPD-1), and soluble CD80 (sCD80). LTA1 is a potent adjuvant that performs similarly to LT, through the recruitment and activation of dendritic cells [26,27]. The IL-12 adjuvant has been widely used in both NHP and human clinical trials [28,29] and promotes differentiation of

naïve CD4+ and CD8+ T cells to Th1 and cytotoxic T lymphocytes (CTLs), respectively. IL-33 has also been shown to augment vaccine immunogenicity in mice [30,31], and works by directly promoting the activity of Th1 cells and CTLs [32,33]. RALDH2 has previously been used as an adjuvant to enhance mucosal vaccine immunogenicity in mice [34], and was included to enhance mucosal vaccine immunogenicity through the conversion of retinaldehyde to retinoic acid, the molecule responsible for inducing the expression of the mucosal homing factors CCR9 and $\alpha 4\beta 7$ on activated lymphocytes [34,35]. Finally, previous studies showed that blocking the PD-1 and CTLA-4 pathways can enhance antigen-specific immunity, reduce immune activation, and reverse immune exhaustion [36,37]. We therefore co-delivered plasmids expressing rhesus sPD-1 and sCD80 to block the interaction of CD8+ T cells expressing PD-1 and CTLA-4 with antigen-presenting cells (APCs) expressing PDL-1 and CD80.

Our results show that the vaccine delivered with the adjuvant combination (MAG + AC) significantly increased IFN-$\gamma$ T cell responses in the blood but not in the gut-associated lymphoid tissue (GALT), and did not achieve a significant improvement in viral control during analytic treatment interruption (ATI, discontinuation of ART) when compared to mock-vaccinated controls. However, a subset of animals among all groups maintained low viremia during ATI, providing an opportunity to investigate immune correlates of this prolonged resistance to viral rebound. Our results show that animals that controlled viral rebound (controllers) during ATI exhibited higher polyfunctional SIV-specific CD8+ T cells in the mesenteric lymph nodes (MLN) and blood. Importantly, increased colonic CD4+ T cells and lower Th17/Treg ratios pre-infection correlated with improved response to ART and lower viral burden during ATI. Together, these data provide new evidence that the state of the mucosal immune system before infection may influence an individual's response to ART and their ability to develop and maintain mucosal and systemic CD8+ T cell responses that can contribute to control of viral rebound during ATI.

## Results

### NHP study design

Rhesus macaques were intravenously infected with SIVΔB670, a highly pathogenic, primary isolate that induces AIDS in most rhesus macaques within 5–17 months of infection [38]. This strain was chosen because the 15% divergence between the Env consensus sequence of the SIVΔB670 inoculum and the SIV/17E-Fr Env sequence in the vaccine mimics therapeutic vaccination of humans infected with diverse variants of a given HIV subtype [15,38,39].

At six weeks post-infection (wpi), animals began ART, consisting of emtricitabine (FTC), tenofovir (PMPA), and Raltegravir, administered daily. Starting at 32 wpi, the macaques in the vaccine groups received a series of 5 DNA immunizations, spaced 4 weeks apart (Fig 1). Prior to initiating therapeutic immunizations, we stratified the animals so that each group had comparable levels of plasma viremia and blood CD4+ T cell counts (S1 Fig) during acute infection and ART to help balance the effects of pre-existing virological and host factors among all groups.

The control group (N = 4) received mock DNA immunizations via PMED consisting of the vaccine plasmid backbone, without SIV antigens or adjuvants. The MAG + LT group (N = 5) received the MAG vaccine, a plasmid that encodes SIV/17E-Fr Gag-Pol-Env and expresses virus-like particles [40] co-delivered with plasmids encoding SIV/17E-Fr p57 Gag and the LT adjuvant we previously showed induces mucosal T cell responses [15,27], also via PMED. A PMED group was included as a comparator because of its previously demonstrated therapeutic efficacy with another DNA vaccine [15]. The MAG vaccine was supplemented with an additional plasmid encoding p57 Gag based on evidence from both human and NHP studies that

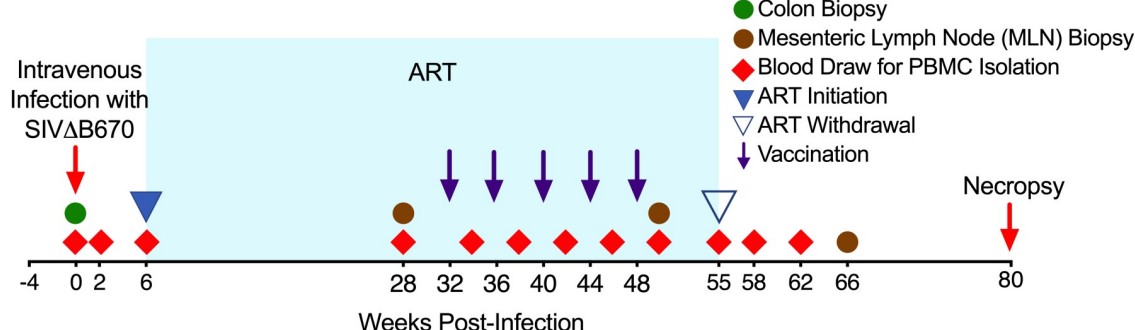

**Fig 1. Therapeutic vaccine study design.** Indian origin rhesus macaques were infected with SIVΔB670 at week 0 (red arrow) and were treated with ART starting at 6 weeks post-infection (wpi). Purple arrows indicate a series of 5 DNA immunizations spaced 1 month apart, occurring between 32 wpi and 48 wpi. At week 55, ART was interrupted to assess the efficacy of the therapeutic vaccine on viral control. Animals were necropsied at 80 wpi or earlier in the presence of AIDS-defining conditions. Red triangles indicate blood draws for PBMC isolation and brown circles indicate MLN biopsies to measure systemic and gut-associated immune responses. Prior to administering therapeutic immunizations, macaques were stratified so that each group had comparable viral loads and CD4 counts prior to and during ART.

Gag-specific T cell responses are crucial for control of viral replication [15,41–46]. Additionally, our previously published therapeutic vaccine study demonstrated that SIVΔB670-infected rhesus macaques vaccinated with a Gag expression plasmid within the context of a multi-epitope SIV DNA vaccine durably controlled viral rebound after stopping ART and consistently demonstrated elevated Gag-specific CD8$^+$ T cell responses [39].

The MAG + AC group (N = 5) received DNA immunizations via intradermal injection followed by electroporation. For the first vaccination, this group of animals received the MAG DNA vaccine co-formulated with plasmids expressing the adjuvants LTA1, IL-12, IL-33, RALDH2. For the subsequent four vaccinations, each macaque in this group received the same vaccine and adjuvants, but also received a plasmid co-expressing sPD-1 and sCD80.

## ART reduces viremia and restores CD4$^+$ T cells in the periphery

To provide potent suppression of viral replication during ART, we employed a combination of three antiretrovirals: the integrase inhibitor, Raltegravir, and the non-nucleoside reverse transcriptase inhibitors FTC and PMPA that were previously shown to effectively suppress SIV infection in rhesus macaques [47]. However, impaired kidney function due to prolonged treatment with PMPA necessitated a switch over to tenofovir disoproxil fumarate (TDF) in four animals: A16144 and A16145 in the MAG + LT group, A16149 in the mock vaccine group, and A16237 in the MAG + AC group. A16144 was switched at 48 wpi, A16145 at 51 wpi, A16149 at 47 wpi, and A16237 at 41 wpi. The switch from FTC to TDF did not cause any notable changes in the level of suppression of viral replication in these animals.

Prior to starting the immunizations at week 32, 12/14 animals responded well to ART, reaching a significant 2.75–4.70 log-fold reductions in median viral loads prior to vaccination (P < 0.0001, S1 Fig). Among these, 9/14 animals exhibited undetectable levels of viremia prior to vaccination (≤30 viral copies per mL of plasma), although low levels of transient viral

replication during ART were detected in all ART-responders. During acute infection, animals experienced a rapid CD4[+] T cell decline that was significantly restored after initiation of ART (P = 0.040, S1 Fig). Two animals did not reach our criteria for effective viral suppression on ART (A16234 in the mock vaccine group and A16236 in the MAG + AC group, S1 Fig), maintaining median viral loads greater than $10^4$ viral copies per mL of plasma. However, these animals still showed a substantial two log-fold decrease in their plasma viremia during ART and did not experience disease progression prior to ATI. Notably, our previous studies indicate that SIVΔB670 appears to be more resistant to ART [15,39] than other SIVs commonly used as inoculums in NHP studies, namely SIVmac251 [48,49] and SIVmac239 [50,51]. This inherent ART resistance likely contributed to persistent viral replication during ART in 2 out of 14 animals. However, the levels of viral suppression on ART in these animals were superior to our previous therapeutic studies utilizing SIVΔB670, including two studies where we reported a significant impact of therapeutic vaccination on viral control in ART responders [15,39]. Our previous therapeutic vaccine studies with SIVΔB670 reported highly variable responses to ART consisting of one or two anti-retroviral drugs, with 40–50% of macaques exhibiting no decrease in viral load during ART [15,39]. Here, using a more potent triple drug combination, 14/14 animals exhibited at least a 3-log decrease in viral load within 4 weeks of ART initiation that was maintained for the duration of ART.

## Therapeutic vaccination with MAG + AC increases SIV-specific IFN-γ T cellular responses

To assess the impact of therapeutic vaccination on antibody responses, levels of SIV gp130-specific IgG were measured by ELISA. Peak antibody titers developed after the first vaccine dose (34 wpi) in the MAG + LT animals and after the second dose (38 wpi) in the MAG + AC group, but antibody responses declined by two weeks after the third dose (Fig 2A). No differences in antibody titer were observed between the MAG + LT and MAG + AC groups.

To determine the impact of the therapeutic vaccines on T cell responses, SIV-specific T cells producing IFN-γ in response to stimulation with peptide pools spanning SIVmac239 p57[Gag] (Gag), gp130 (Env), Pol, Vif, Vpr, Rev, Nef, and Tat were measured by ELISpot. Although the sequences are not a precise match for the SIVΔB670 inoculum or the SIV/17E-Fr strain used to construct the DNA vaccine, the genetic diversity between the three viruses resembles the intraclade diversity observed for HIV-1 [38] and thus are relevant in the context of human HIV-1 infection. However, we cannot rule out the possibility that some vaccine or challenge strain-specific IFN-γ responses were not detected in our T cell assays due to these differences. The median SIV-specific IFN-γ response in the MAG + LT group peaked after the first vaccine dose, but declined afterwards despite additional vaccinations (Fig 2B). In contrast, the median IFN-γ response in the MAG + AC animals increased with each vaccine dose up to the third dose (Fig 2B), resulting in significantly higher responses compared to the MAG + LT group two weeks after the fifth and final DNA vaccine dose (P = 0.022, Fig 2C). This is consistent with our previous study, where a PMED-delivered therapeutic DNA vaccine administered over six doses induced peak IFN-γ responses after the third dose [15]. It was not possible to distinguish between virus-stimulated and vaccine-induced immune responses from these experiments. However, SIV-specific IFN-γ responses were mostly undetectable prior to vaccination (32 wpi), suggesting that most animals did not have these responses prior to vaccination. We thereby conclude that the increase in responses detected post-vaccination is due to the vaccine inducing de novo T cell responses or boosting virus-primed responses, rather than ongoing viral replication.

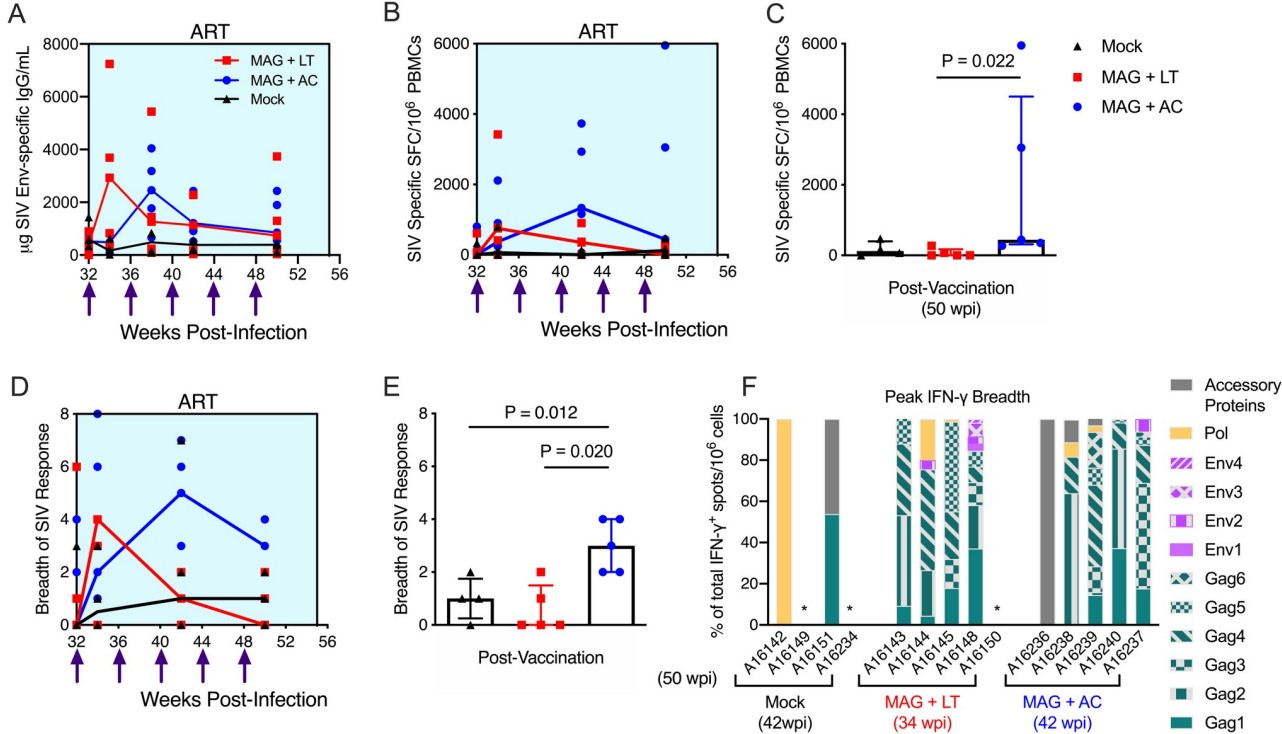

**Fig 2. DNA vaccination increases Env-specific antibody responses and total SIV-specific IFN-γ T cell responses during ART. (A)** The magnitude of the SIV Env-specific IgG response in the plasma was measured by ELISA, using SIV gp130 as the capture antigen. Shown are medians with individual animals' data layered over. **(B-C)** PBMCs were stimulated with Gag, Env, Pol, Vif, Vpr, Rev, Nef, and Tat peptides to quantify the SIV-specific IFN-γ response. Samples were considered positive if peptide-specific responses were at least twice that of the negative control plus at least 0.01% after background (DMSO) subtraction. **(B)** Shown are medians of the cumulative (sum of response against all peptides) IFN-γ response, with data from individual animals layered over. **(C)** The cumulative SIV-specific IFN-γ response is shown after 5 vaccinations (50 wpi). Shown are medians and interquartile ranges with individual responses layered over each bar. **(D)** The breadth of the SIV-specific IFN-γ response is the number of peptide pools with a positive IFN-γ response. The cumulative breadth of the SIV-specific IFN-γ response post-vaccination (50 wpi) is shown, with medians and interquartile ranges of each group depicted, with individual animals' data layered over each bar. **(E)** The cumulative breadth of the SIV-specific IFN-γ response post-vaccination (50 wpi) is shown as medians and interquartile ranges with individual responses layered over each bar. **(C, E)** A Dunn's multiple comparisons test was used when making multiple comparisons between vaccine groups and the mock group. Results are considered significant if P ≤ 0.05. Shown are the medians and individual responses layered over each timepoint. **(F)** The percent of the IFN-γ response specific for Gag, Env and accessory proteins was calculated from the cumulative IFN-γ response at peak breadth (34 wpi for MAG + LT, 42 wpi for MAG + AC and Mock).

The breadth of the IFN-γ T cell response was also measured by IFN-γ ELISpot, where T cell responses were mapped against pools of overlapping peptides (15-mers overlapping by 11 amino acids) spanning SIVmac239 Gag, Env, Pol, Vif, Vpr, Rev, Nef, and Tat, and defined as the number of peptide pools eliciting an IFN-γ response. In the MAG + LT group, the breadth of the IFN-γ T cell response peaked after the first vaccine dose (34 wpi) and subsequently declined, while the breadth of the MAG + AC group peaked two weeks after the third dose (42 wpi, Fig 2D). Following the final vaccine dose (50 wpi), the MAG + AC vaccine group sustained significantly greater breadth in T cell responses than both the MAG + LT group (P = 0.020, Fig 2E) and the mock-vaccinated group (P = 0.012, Fig 2E). Interestingly, during the peak IFN-γ T cell response post-vaccination, the response was predominantly directed towards Gag in both vaccine groups, with up to 100% of the IFN-γ response targeting Gag sequences in both the MAG + LT and MAG + AC groups (Fig 2F). Responses to Pol were detected in two out of five animals in both the MAG + LT and MAG + AC groups, accounting for up to 20% of the total IFN-γ response (Fig 2F). Env-specific responses were identified in two out of five animals in the MAG + LT group and one out of five animals in the MAG + AC

group, with up to 15.5% of the total IFN-γ response targeting Env. SIV-specific IFN-γ responses were not observed in one animal in the MAG + LT group (A16150), while one animal in the MAG + AC group (A16236) only exhibited IFN-γ responses to accessory proteins, indicating vaccination had no effect on the SIV-specific IFN-γ response in 1/5 animals in each vaccine group. In contrast, IFN-γ responses in the mock-vaccinated group were only detected in two out of four animals and predominantly targeted either Pol or accessory proteins (Vif, Rev and Nef), with Gag-specific responses detected in only one animal (Fig 2F).

Env- and Gag-specific T cell responses were further characterized in the PBMC and MLN for effector functions, including secretion of IFN-γ, TNF-α, and IL-2, and co-expression of CD107a/Granzyme B as markers of cytolytic function by intracellular cytokine staining (ICS) and flow cytometry. In contrast to the IFN-γ ELISpot assay, where we analyzed the SIV-specific IFN-γ response as a whole in bulk PBMCs, here we assessed CD4$^+$ and CD8$^+$ T cell effector functions separately. There were no significant differences between groups in these functions in either the PBMC or MLN (S2–S5 Figs), although the MAG + AC group demonstrated a trend towards higher cumulative IFN-γ$^+$ CD4$^+$ and CD8$^+$ T cells in the PBMC (S6 Fig). This trend was not statistically significant, likely due to the significant differences in the populations of cells analyzed by each assay. However, the ICS analysis affirms our ELISpot results are representative of the overall IFN-γ$^+$ T cell response.

To further characterize the SIV-specific CD8$^+$ T cell response post-vaccination, we assessed the magnitude of the polyfunctional CD8$^+$ T cell response. Here, we define polyfunctionality as the frequency of T cells specific for either Gag or Env and expressing any three or more of the cytokines IFN-γ, TNFα, IL-2, and/or co-expressing the cytolytic markers CD107a/Granzyme B. Three out of five animals in the MAG + AC group exhibited increases in SIV-specific polyfunctional CD8$^+$ T cells post-vaccination in both PBMC and MLN, whereas the frequency of polyfunctional CD8$^+$ T cells in both the MAG + LT and mock-vaccinated controls remained nearly undetectable. However, these results were not statistically significant (S7 Fig).

## Impact of therapeutic vaccination on protection from viral rebound during analytical treatment interruption (ATI)

To determine if therapeutic vaccination improved viral control, ATI was initiated three weeks after the final vaccine dose (55 wpi), and viral loads were monitored for 6 months. Containment of median viral loads at or below $10^3$ copies/mL of plasma during ATI was chosen as the primary criterion for therapeutic efficacy based on previous studies showing that rhesus macaques infected with SIVΔB670 that maintained viral loads at or below this level in the absence of ART consistently exhibit long term (>1 year) protection from progression to AIDS [40]. During ATI, 60% (3/5) of animals in the MAG + AC group maintained viremia below $10^3$ RNA copies/mL of plasma for 6 months and sustained CD4 counts above 50% of pre-infection levels, whereas only one animal in the MAG + LT group (1/5, 20%) and one in the control group (1/4, 25%) exhibited similar viral control and protection from disease progression (Fig 3A–3C). However, these differences between groups were not statistically significant. Vaccinated and control animals that exhibited immediate viral rebound during ATI also showed significant CD4$^+$ T cell decline during ATI (S8 Fig). Overall, there was no significant difference in protection from viral rebound or median viral loads during ATI between the three groups (Fig 3D), an outcome that is likely due to the small group sizes coupled with variability in response to ART across all three groups.

In our previous therapeutic vaccine study, we excluded animals with high levels of persistent viral replication on ART from our analyses and observed a significant impact of therapeutic vaccination only in animals categorized as ART responders (Median viral loads < 5 x 10$^4$

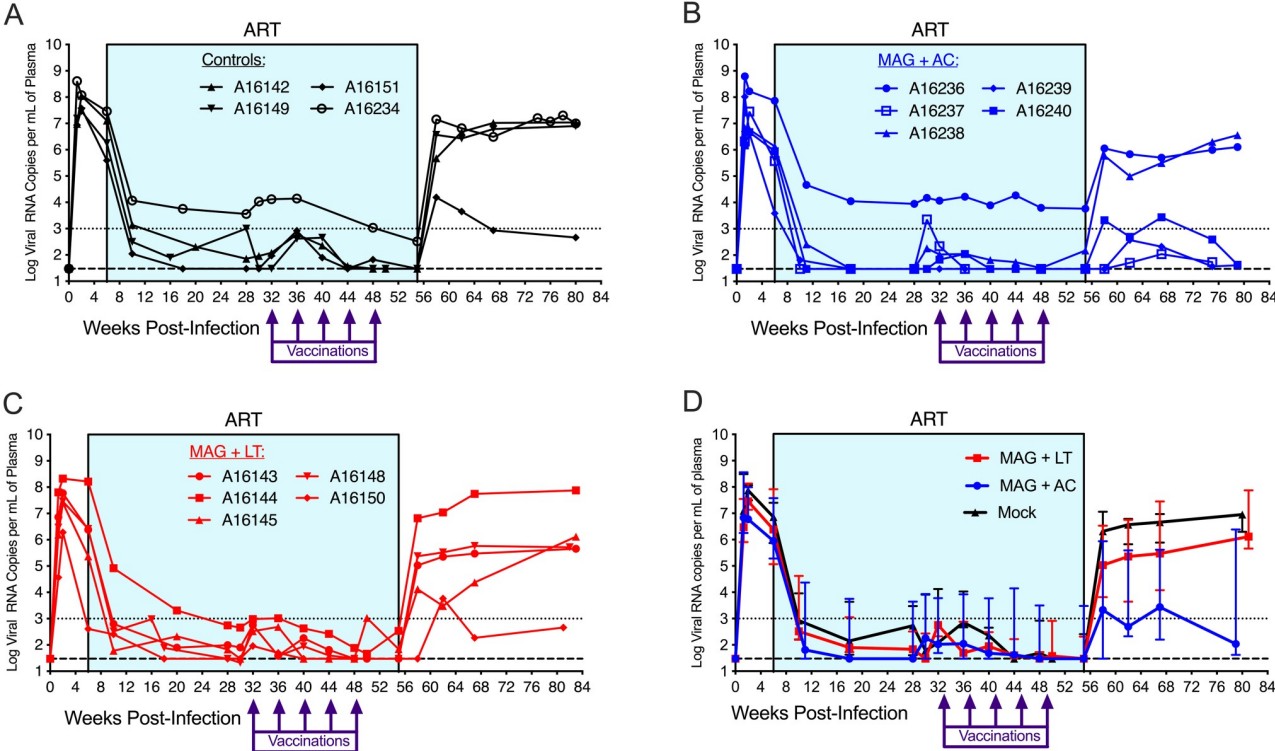

**Fig 3. Three out of five animals in the MAG + AC group control virus replication during ATI. (A-C)** Plasma viral RNA levels were quantified using RT-q-PCR, with a limit of detection of 30 viral RNA copies per 1 mL of plasma, as indicated by the dashed line. **(D)** Shown are the median viral load and interquartile ranges for each treatment group. The dotted line indicates the threshold for control of virus replication, based on previous studies using SIVΔB670.

RNA copies/mL of plasma during ART) [15]. Although the animals in this study demonstrated a wide range of virologic suppression on ART, all were considered ART responders based on the criteria used in our two previous therapeutic vaccine studies utilizing SIVΔB670. However, to address the possibility that the two animals with incomplete viral suppression on ART could be skewing our results, we added a supplemental figure with A16234 from the mock-vaccinated group and A16236 from the MAG + AC group excluded (S9 Fig). This demonstrates that excluding these animals from analysis of vaccine efficacy did not impact our conclusions, likely due to the small group sizes.

## Viral control during ATI is associated with increased Gag-specific CD8[+] T cells in MLN and PBMC

The variability in viral rebound and viremia during ATI among animals in this study enabled further study of immune correlates of viral control. Altogether, there were 5 viral controllers and 9 non-controllers, defined as animals that maintained median viremia at or below $10^3$ copies/mL of plasma or greater than $10^3$ copies/mL of plasma, respectively, for 5 months after stopping ART (Fig 4A). Viral burden during ATI was significantly different between controllers and non-controllers (P = 0.0010, Fig 4B) Viral burden during ATI was significantly different between controllers and non-controllers (P = 0.0010, Fig 4B). Certain MHC and TRIM5 genetics have been associated with improved viral control in SIV infected macaques [52–57], however, we observed no association between these MHC or TRIM5 alleles and the control of

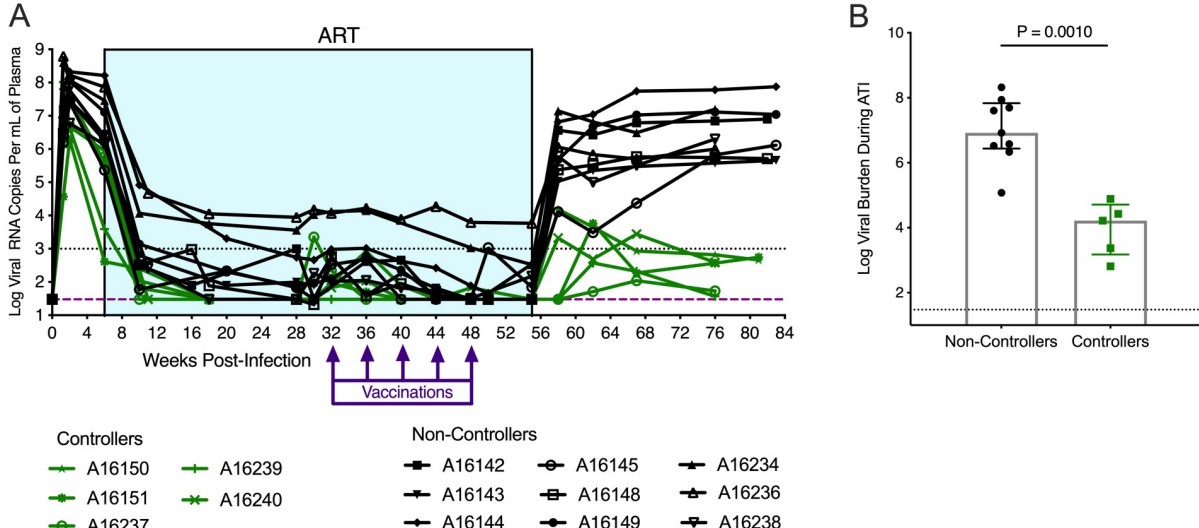

**Fig 4. Five controllers maintained significantly lower viral burden during ATI compared to nine non-controllers.** (A) Plasma viral RNA levels were quantified using RT-q-PCR, with a limit of detection of 30 viral RNA copies per 1 mL of plasma, as indicated by the dashed line. The dotted line denotes the threshold for control of virus replication, based on previous studies using SIVΔB670. Controllers were defined as animals that maintained a median viremia at or below 1000 viral RNA copies per 1 mL of plasma for 5 months post-ART. Non-controllers were defined as animals with a median viremia that exceeded 1000 viral RNA copies per 1mL of plasma for 5 months post-ART. (B) Viral burden during ATI was calculated as the area under the curve of each animal's viral load from 55 wpi to 76 wpi, shown are the median viral burden and interquartile ranges. Statistics were calculated using a Mann-Whitney t-test. Results are considered significant if P ≤ 0.05.

viral rebound (S1 Table) indicating these genotypes did not likely influence viral burden during ATI in this study.

To determine immune correlates of viral control, we first compared Env-specific IgG titers in controllers and non-controllers, and found that non-controllers exhibited a consistent trend towards higher titers of Env-specific IgG both during ART treatment and during ATI (S10 Fig), likely due to higher levels of ongoing virus replication.

Next, we compared frequencies of Gag-specific CD4+ and CD8+ T cells expressing IFN-γ, TNFα, IL-2, and/or co-expressing the cytolytic markers CD107a/Granzyme B as detected by flow cytometry in the controllers and non-controllers. Most therapeutic vaccine studies in NHP are limited to investigating immune correlates in the peripheral blood. Here, we sought to establish the role of T cell responses in both the blood and the GALT on viral control. Immune responses in PBMC and MLN were compared prior to ATI (50 wpi) and after viral setpoint was established (62 wpi for PBMC and 66 wpi for MLN). We observed no differences between controllers and non-controllers in terms of Gag-specific CD4+ T cell responses in PBMC or MLN (S11 Fig). However, controllers demonstrated a trend towards higher frequencies of Gag-specific TNFα+ CD8+ T cells in PBMC prior to ATI (50 wpi, Fig 5A, P = 0.056) and exhibited significantly higher frequencies of Gag-specific IFN-γ+ CD8+ T cells in PBMC during ATI (62 wpi, Fig 5B, P = 0.0080). In the MLN, controllers had significantly higher frequencies of Gag-specific IL-2+ CD8+ T cells pre-ATI (50 wpi, Fig 5C, P = 0.037), although this was not sustained during ATI (66 wpi). Importantly, higher frequencies of Gag-specific IFN-γ+ and TNFα+ CD8+ T cell responses in PBMC and Gag-specific IL-2+ CD8+ T cell responses in the MLN significantly correlated with lower viral burden during ATI (Fig 5D–5F), suggesting that SIV Gag-specific T cell responses in both the periphery and GALT contribute to the improved control of virus replication in controllers. Although we cannot completely rule out a

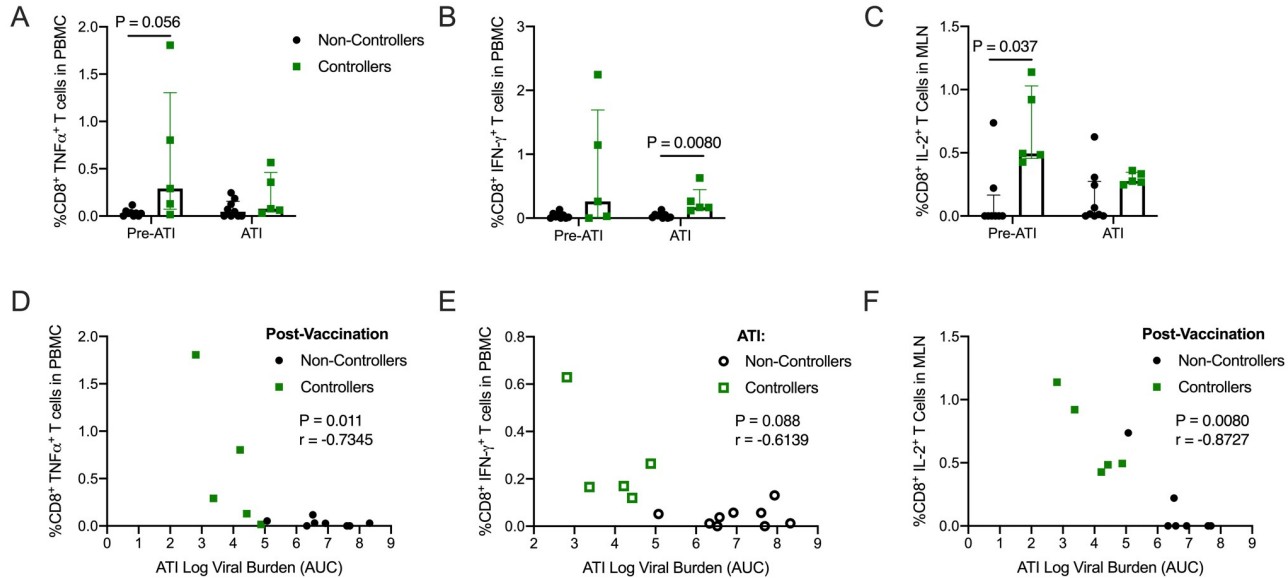

**Fig 5. Controllers have higher SIV Gag-specific CD8+ T cell responses in PBMC and MLN post-vaccination and during ATI. (A-C)** PBMCs and MLNs were thawed and stimulated with Gag peptides, and expression of cytokines was quantified using intracellular cytokine staining. Shown are the medians and interquartile ranges of the SIV Gag-specific CD8+ T cell responses of controllers and non-controllers, with individual responses layered over each bar at a post-vaccination timepoint (50 wpi) and during ATI (62 wpi for PBMC and 66 wpi for MLN). Statistical differences between controllers and non-controllers at each timepoint were calculated using a Mann-Whitney t-test. Benjamini-Hochberg adjusted P values are shown, results are considered significant if P ≤ 0.05. **(D)** The SIV Gag-specific TNFα CD8+ T cell responses in PBMC at 50 wpi negatively correlated with the viral burden measured as area under the curve (AUC) during ATI. **(E)** The SIV Gag-specific IFN-γ CD8+ T cell responses in PBMC at 50 wpi negatively correlated with the viral burden measured as area under the curve (AUC) during ATI. **(F)** The SIV Gag-specific IL-2 CD8+ T cell responses in MLN at 62 wpi negatively correlated with the viral burden measured as area under the curve (AUC) during ATI. The P and r values shown were calculated using a Spearman rank correlation test. Benjamini-Hochberg adjusted P values are shown, results are considered significant if P ≤ 0.05.

role for strain-specific neutralizing or non-neutralizing antibodies, these results suggest viral control during ATI was likely mediated, at least in part, by CD8+ T cell responses.

## Viral control during ATI is associated with increased polyfunctionality in MLN and PBMC

To further elucidate the role of SIV-specific CD8+ T cell responses in viral control, we next compared the magnitude of the polyfunctional CD8+ T cell response, as defined by the frequency of T cells specific for either Gag or Env and expressing any three or more of the cytokines IFN-γ, TNFα, IL-2, and/or co-expressing the cytolytic markers CD107a/Granzyme B. Prior to ATI (50 wpi), higher frequencies of polyfunctional CD8+ T cells expressing three effector functions in the MLN (P = 0.016, Fig 6A) and PBMC (P = 0.013, Fig 6B) correlated with lower viral burden during ATI. Interestingly, during ATI (62 wpi in PBMC and 66 wpi in MLN), the significant correlation between higher frequencies of polyfunctional CD8+ T cells in MLN and lower ATI viral burden (P = 0.015, Fig 6C) persisted, whereas in the PBMC, the correlation fell below statistical significance (P = 0.12, Fig 6D). These results indicate that while polyfunctional CD8+ T cell responses in the periphery likely played an integral part in controlling viral recrudescence during ATI, responses in the GALT may be more critical for sustained containment of viral rebound. Notably, there are no differences between controllers and non-controllers in the frequencies of CD8+ T cells expressing one or two effector functions (non-polyfunctional responses) before or during ATI (S13 Fig). Although non-polyfunctional responses make up the majority of the SIV-specific CD8+ T cell response, these results demonstrate that polyfunctional CD8+ T cells likely play a greater role in controlling viral replication.

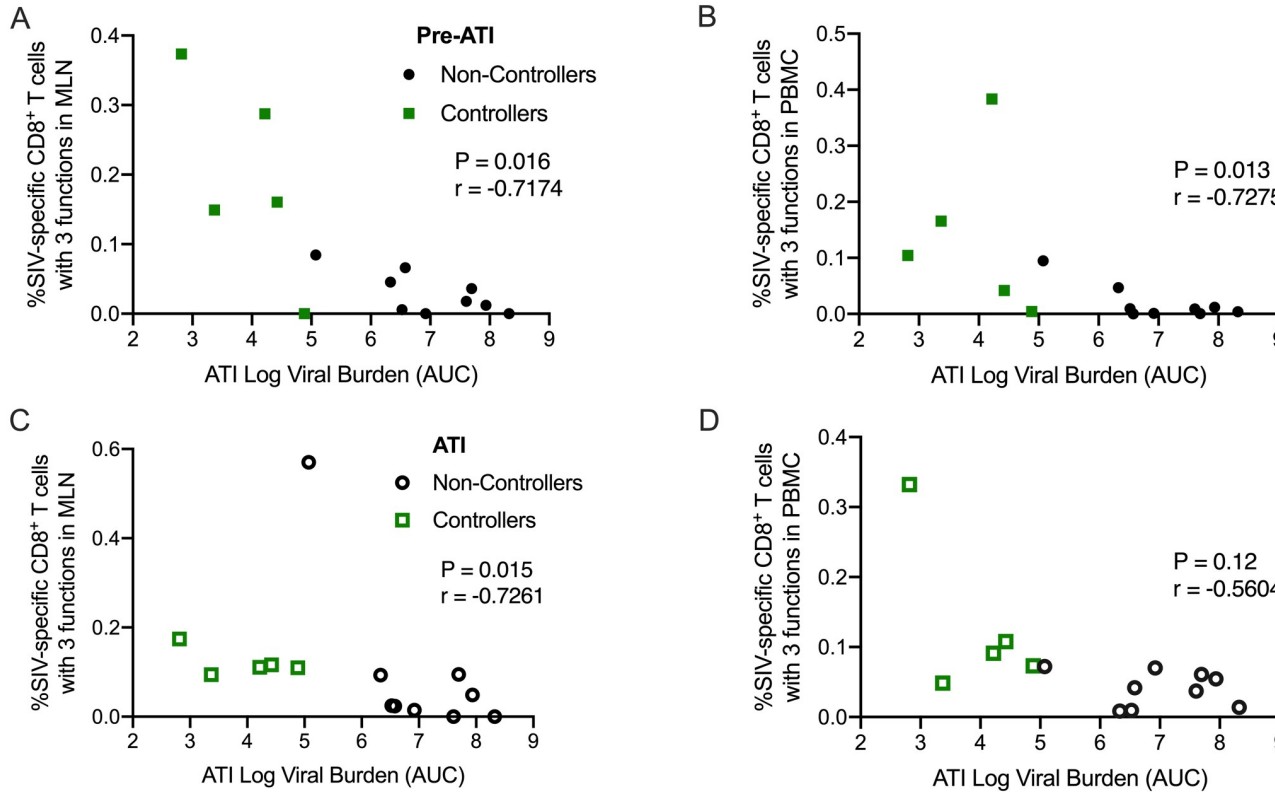

**Fig 6. Increased frequencies of polyfunctional CD8+ T cells in MLN and PBMC of controllers. (A-D)** The polyfunctional SIV-specific CD8+ T cell responses in MLN and PBMC post-vaccination and during ATI negatively correlated with ATI viral burden. The P and r values shown were calculated using a Spearman rank correlation test. Benjamini-Hochberg adjusted P values are shown, and results are considered significant if P ≤ 0.05.

The importance of T cell polyfunctionality in the blood in viral control during ATI is well-established [15,58,59]. Our data extends these findings and suggests that polyfunctional T cell responses in the GALT as well as the PBMC are likely key for achieving durable immune control of viral replication.

## ART responsiveness and pre-infection populations of mucosal CD4+ T cells predict control of viremia during ATI

A significant variable in our study is the wide range of acute viral loads and response to ART that occurred prior to initiating therapeutic vaccinations. To determine if these variables influenced viral control during ATI, we compared the viral loads of controllers to non-controllers during acute infection (0–6 wpi) and during ART but prior to vaccinations (6–32 wpi). Controllers demonstrated a trend towards lower viral burden during acute infection (P = 0.11, Fig 7A) and significantly lower viral burden while on ART (P = 0.014, Fig 7A) when compared to non-controllers.

Importantly, lower acute viral burden was associated with lower residual viral replication during ART (P = 0.0078, Fig 7B), and lower viral burden on ART was strongly correlated with better control of viral replication during ATI (P = 0.00040, Fig 7C). Together, these data suggest that pre-infection host immune parameters may have influenced the extent of acute viral replication and subsequently affected the extent of residual viral replication on ART, or ART efficacy. Persistent viral replication on ART in turn may have influenced the ability of each animal to develop and maintain polyfunctional CD8+ T cell responses in the blood and GALT

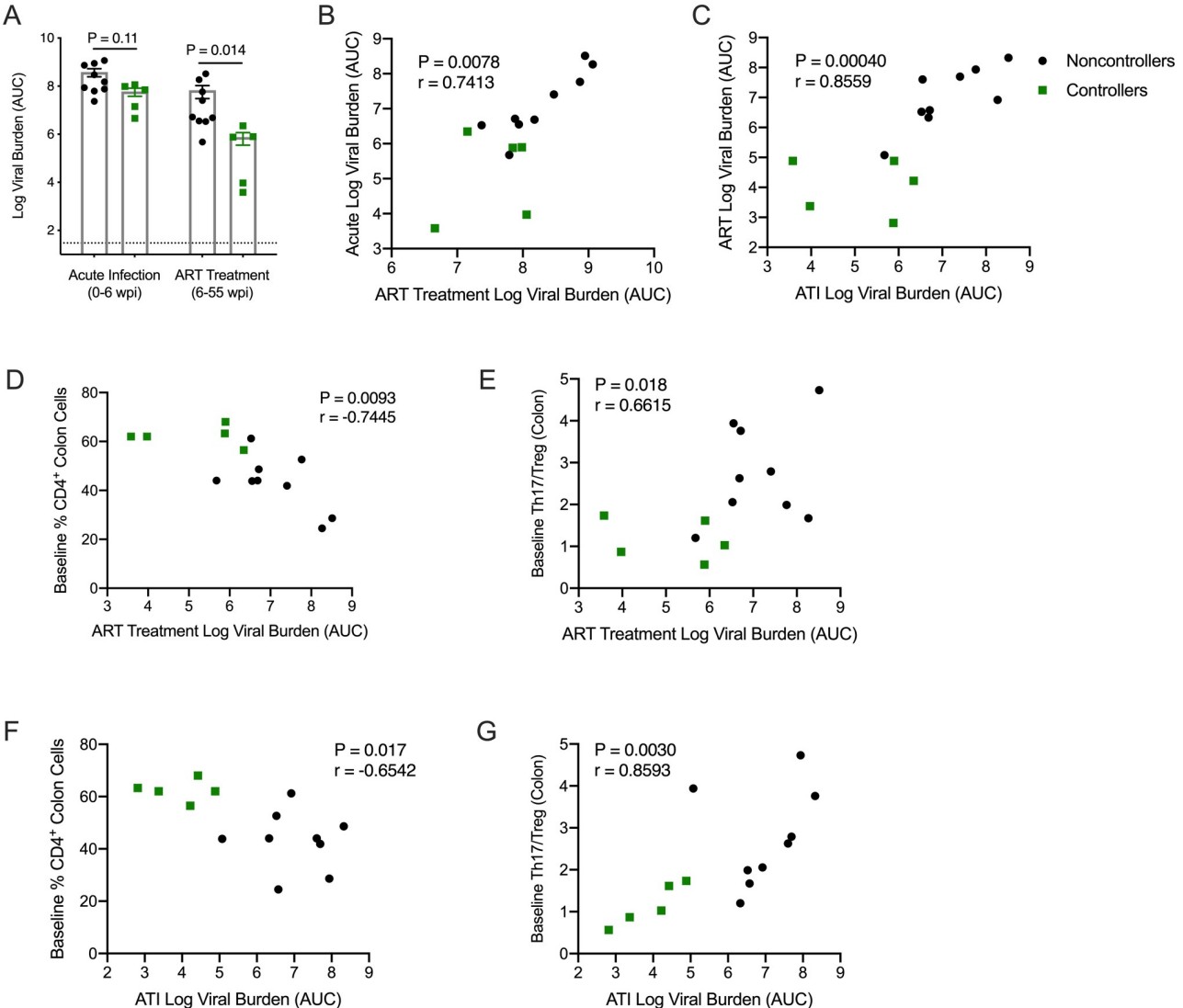

**Fig 7. Lower viral replication during acute infection correlates with improved ART responsiveness that in turn is associated with better control of viral rebound during ATI.** Improved virological response to ART and better control of viral rebound are associated with the relative populations of mucosal CD4+ T cells and Th17/Treg cells before infection. **(A-C)** Viral loads were measured via RT-q-PCR and viral burden was calculated as the area under the curve of animals' viral loads. **(A)** Statistics were calculated using a Mann-Whitney t-test. Benjamini-Hochberg adjusted P values are shown. Results are considered significant if P ≤ 0.05. **(B-C)** Acute log viral burden (0–6 wpi) correlated with log viral burden during ART treatment (6–32 wpi). ART log viral burden in turn correlated with log viral burden during ATI (55 wpi– 76 wpi). **(D)** The frequency of CD4+ T cells in the colon at baseline (0 wpi) correlated with viral burden during ART. **(E)** The ratio of Th17 and Treg cells at baseline correlated with viral burden during ART. **(F)** The frequency of CD4+ T cells in the colon at baseline (0 wpi) correlated with viral burden during ATI. **(G)** The ratio of Th17 and Treg cells at baseline correlated with viral burden during ATI. **(B-G)** A Spearman rank correlation test was used to determine P and r values. Shown are Benjamini-Hochberg adjusted P values, results are considered significant if P ≤ 0.05.

during ART that significantly correlated with control of viral replication during ATI (Fig 6A–6D).

To investigate this theory, we assessed a number of baseline immune factors, including CD4+ T cells, T helper 17 (Th17) and T regulatory (Treg) cells in the colon, that we hypothesized could influence acute viral infection and potentially affect the virological response to ART and outcome of ATI. In particular, depletion of Th17 cells in the gut contributes to immune activation and disease progression [60,61]. The role of Treg cells in HIV pathogenesis

is not as well-characterized, but they could contribute to viral replication by suppressing HIV-specific CD8+ T cell activity [62] or conversely, slow disease progression by decreasing chronic immune activation [63]. In the colon, neither baseline frequencies of Th17 (P = 0.22) nor Treg (P = 0.14) cells correlated with viral burden during ATI (S14 Fig). However, upon closer examination, we observed a correlation between greater CD4+ T cell frequencies (P = 0.0093, Fig 7D) and lower Th17/Treg ratios (P = 0.018, Fig 7E) and improved virological response to ART. Furthermore, higher frequencies of colonic CD4+ T cells and lower Th17/Treg ratios at baseline were predictive of lower viral burden during ATI (P = 0.017, Fig 7F and P = 0.0030, Fig 7G). Collectively, these data suggest that during the early stages of infection, the relative proportions of mucosal T cell subsets play an important role in maintenance of gut homeostasis and prevention of immune dysfunction, and may influence responsiveness to ART, therapeutic vaccination and/or control of viremia during ATI.

## Discussion

The primary goals of this study were to assess the immunogenicity and efficacy of a multi-antigen DNA vaccine (MAG) delivered with a novel genetic adjuvant combination (AC), and to determine what host factors could be influencing the effects of therapeutic vaccination. To this end, we reported that animals in the multi-antigen vaccine and adjuvant combination group (MAG + AC) exhibited significant increases in the breadth of IFN-γ T cell responses when compared to the vaccine group that received MAG with only a single adjuvant (MAG + LT) and mock-vaccinated controls, an outcome that is consistent with other vaccine trials in NHP and mice using the IL-12 and IL-33 adjuvants [28,31,64]. However, the RALDH2 adjuvant did not enhance vaccine immunogenicity in the GALT as shown in mice [34], possibly due to SIV infection causing significant mucosal immune dysfunction in the NHP that may have interfered with the effects of this adjuvant. Additionally, we observed that therapeutic vaccination only transiently boosted the magnitude of the IFN-γ T cell responses and Env-specific antibody responses in both the MAG + AC and MAG + LT groups. This is similar to what we reported in our previous study [15] and may be the result of increases in regulatory immune responses, such as myeloid derived suppressor cells (MDSCs), that occur during acute infection and persist despite ART treatment. Three out of five MAG + AC animals controlled viral rebound (60%), although this outcome was not statistically different from the controls. The lack of significant differences between the vaccine and control groups may be due to the small group sizes and the variability in the response to ART within each group. This is consistent with a previous study where we reported significant viral control during ATI following therapeutic immunization with a PMED DNA vaccine, but only in a subset of animals that responded well to ART [15]. However, since the three controllers in the MAG + AC group exhibited a better virological response to ART prior to vaccination than the two non-controllers, we cannot conclusively determine whether improved viral control in a subset of animals in this group was due to the vaccine or pre-existing intrinsic host factors that influenced their response to ART.

Overall, five animals from all groups controlled viral rebound and were protected from progression to AIDS, in contrast to nine animals that exhibited immediate viral rebound during ATI. This variability in viral control during ATI enabled further analysis of immune correlates of protection from viral rebound. Notably, we found that increased expression of IL-2 and higher frequencies of polyfunctional, SIV-specific CD8+ T cells in the MLN prior to stopping ART were associated with lower viral burden during ATI. We also observed that CD8+ T polyfunctionality and expression of IFN-γ and TNFα in PBMC were associated with lower viral burden during ATI. The disparity between our observations in the MLN and PBMC is

consistent with previous studies reporting significant variations between CD8[+] T cells in blood and lymphoid tissues in SIV-infected rhesus macaques [65], and high frequencies of tissue-resident HIV-specific CD8[+] T cells in elite controllers that are distinct in their functionality from CD8[+] T cell responses in the blood [66]. It is possible that these differences in cytokine expression and polyfunctionality arose because CD8[+] T cells were primed *in situ* and did not migrate between peripheral compartments and the GALT during chronic infection. Alternatively, these cells may have been primed in peripheral lymph nodes and trafficked to the GALT, subsequently forming highly stable tissue-resident populations [65]. Collectively, these data provide evidence that polyfunctional CD8[+] T cell responses in not just the blood, but also the GALT and gut mucosa are important for controlling reactivating virus in mucosal reservoirs. Therefore, future therapeutic vaccine efforts should be directed towards inducing polyfunctional mucosal responses.

While our data clearly show a protective role for mucosal and systemic polyfunctional CD8[+] T cell responses, the precise mechanisms underlying viral control or recrudescence during ATI are still unclear. Although ART significantly reduced viral burden in these animals, low levels of detectable SIV viremia (median viral loads of $<10^3$ viral copies per mL of plasma) persisted in most animals. Importantly, we found controllers exhibited significantly lower viral loads than non-controllers during ART, and this correlated with lower viral burden during ATI. This is accordant with other NHP studies showing correlations between lower acute SIV viremia and lower SIV viral loads during ATI [67]. Our results extend these findings and show that even low levels of persistent viral replication during ART can affect control of viremia during ATI. Although quantifying viral diversity was outside the scope of this study, persistent viral replication during ART likely increased viral diversity, especially in non-controllers relative to controllers. Higher viral diversity increases the likelihood of immune escape from vaccine-induced immune responses and may have contributed to failure to control viral replication during ATI in some animals. Additional studies are needed to determine how incomplete suppression of viral replication during ART impacted viral diversity in this study and its impact on therapeutic vaccine efficacy.

Another factor that could have influenced post-ART viral rebound is the size of the latent reservoir. Previous studies showed levels of proviral DNA in the GALT correlate with time to viral rebound [68], and increased proviral DNA in PBMC correlates with higher viral loads during ATI [69]. It is therefore possible that the controllers in this study had reduced proviral reservoirs that contributed to improved control of virus replication. Further experiments are needed to quantify cell-associated viral DNA in the GALT and PBMC and to determine how the latent reservoir may have affected virologic outcome during ATI.

To further elucidate mechanisms underlying the response to ART and ultimately viral control during ATI, we investigated a possible role for intrinsic baseline immune factors and found that pre-infection immune responses in the GALT may influence acute viral replication, ART responsiveness, and the ability to control viremia during ATI. Specifically, we found correlations between higher pre-infection frequencies of colonic CD4[+] T cells and lower viral burden during ATI, a finding that is consistent with previous studies in humans, where HIV-specific CD4[+] T cell proliferation and function during early infection were associated with viral control and slower disease progression in the absence of ART [70,71]. Together, these data indicate that robust CD4[+] T cell responses in the mucosa may play a key role in mitigating acute viral infection. We did not detect an association between baseline frequencies of Treg cells or Th17 cells and viral control during ATI. However, when considered together, we observed that higher ratios of Th17 to Treg cells pre-infection correlated with higher viral burden during ATI. This result is consistent with a previous NHP study from this lab, where a higher Th17/Treg ratio measured prior to SIV infection predicted higher acute viremia [72].

When considered together, this suggests that the balance between Th17 and Treg cells during the earliest stages of infection impacts acute viremia and the virological response to ART. Furthermore, the disruption of mucosal Th17 and Treg homeostasis, coupled with persistent, low-level viremia during ART, likely compromised the non-controllers' ability to develop or maintain polyfunctional CD8+ T cell responses in the blood and GALT and their subsequent failure to control virus during ATI.

In summary, the results reported here further our understanding of how robust, polyfunctional CD8+ T cell responses in the GALT contribute to control of SIV replication, and highlight the need for therapeutic HIV vaccines that can induce mucosal immunity. Furthermore, our observations provide new insight into the importance of effective ART as a crucial component of therapeutic interventions. Finally, the data presented here show that mucosal CD4+ T cell homeostasis prior to infection can have far-reaching effects on an individual's ability to control viral rebound during ATI, suggesting that monitoring and maintaining the Th17/Treg ratio may be key to the success of immune-based HIV cure strategies. Altogether, these results underscore the need for therapeutic vaccines and adjuvants capable of enhancing polyfunctional CD8+ T cell responses in both the GALT and the periphery and emphasize the need for additional studies to more clearly define the role of inherent host factors in shaping the response to ART and therapeutic interventions.

## Materials and methods

### Ethics statement and animal care

Fourteen male, adult rhesus macaques (*Macaca mulatta*) of Indian origin were used for this study. These animals were housed at the Washington National Primate Research Center (WaNPRC), which is accredited by the American Association for the Accreditation of Laboratory Animal Care International (AAALAC). At the WaNPRC, animals received the highest standard of care from a team of highly trained, experienced animal technicians, veterinarians, and animal behavior specialists.

Animals were singly housed in stainless-steel cages with a minimum of 6.0sq ft of floor space. In animal housing areas, humidity was maintained between 30–70% while temperature was kept at a range of 72–82˚F. Paired animals were kept in adjacent cages to allow for grooming contact. Waste pans were cleaned daily, while cages, racks, and accessories were sanitized in mechanical cage washers at least once every two weeks. Animals were fed a commercial monkey chow, supplemented daily with fruits and vegetables, and drinking water was available at all times. Environmental enrichment was provided throughout the duration of the study, including grooming contact, perches, toys, foraging experiences, and food enrichment. Animal care staff monitored the health and well-being of all animals on a daily basis. All biopsies, surgeries, and blood draws were performed under ketamine anesthesia (10 mg/kg) and any continuous discomfort or pain was alleviated at the discretion of the veterinary staff to minimize suffering.

Rhesus macaques were challenged intravenously with 100 $TCID_{50}$ of cryopreserved SIVΔB670, diluted in 1 milliliter of RPMI. Following SIV infection, animals were monitored for changes in weight, complete blood counts, blood CD4+ T cell count, and clinical signs of opportunistic infections. Simian AIDS was defined according to WaNPRC guidelines, namely: weight loss exceeding 15%, anemia (sustained hematocrit <15%), CD4+ T cell decline to less than 200 cells per microliter, and presence of opportunistic infections. All four criteria were evaluated in each animal at least monthly and at each specimen collection time point, or more frequently if two or more signs of disease progression were present. None of the animals required euthanasia prior to their experimental endpoint at 80 weeks post-infection.

At the experimental endpoint, euthanasia was performed by administration of Euthasol®
(Virbac Corp., Houston, TX) while the animal was under deep anesthesia, in accordance with
the 2007 American Veterinary Medical Association Guidelines on Euthanasia.

The University of Washington's Institutional Animal Care and Use Committee (IACUC)
approved all experiments in these macaques (IACUC Protocol # 4266–04).

## MHC-I and TRIM5 typing

All macaques were major histocompatibility class I (MHC-1) typed for *Mamu* alleles (A*01,
A*02, A*08, A*11, B*01, B*08, B*17, and B*29). DNA was extracted using the Roche© MagNA
Pure™ system (Roche, catalog number: 06541089001) and analyzed via PCR by Dr. David Watkins and the MHC Genotyping Service at the University of Miami, as previously described
[52,53]. All animals were also tested for TRIM5 haplotypes, including TFP, Q, and CypA, by
PCR of genomic DNA by Dr. David O'Connor at the Wisconsin National Primate Research
Center (WNPRC).

## Quantification of plasma viral load and complete blood counts (CBCs) and serum chemistries

The Virology Core at the WaNPRC, led by Dr. Shiu-Lok Hu and Dr. Patricia Firpo, quantified
viral RNA in the plasma of SIVΔB670-infected animals using a real-time quantitative PCR
(RT-q-PCR) assay. The Virology Core also determined complete blood counts, using a Beckman Coulter® AC*T™ 5diff hematology analyzer (Beckman Coulter) as described previously
[73].

## Antiretroviral therapy

All SIV-infected animals were treated with a combination of 3 antiretroviral therapies: 9-
(2-Phosphoryl-methoxypropyly) adenine (PMPA or tenofovir; provided by Gilead Sciences)
was resuspended in phosphate-buffered saline (PBS) at120 mg/mL. To completely dissolve the
PMPA, 1 molar NaOH was added until the pH reached 7.4–7.8. The solution was then filter
purified, injected into sterile glass vials, and stored at -20˚C. PMPA was administered subcutaneously in a once-daily dose of 20 milligrams per kilogram (mg/kg) of animal weight. 2',3'-
dideoxy-5-fluoro-3'-thiacytidine (FTC or emtricitabine, provided by Gilead Sciences) was
resuspended in PBS at 120 mg/mL. The mixture was heated at 37˚C with constant stirring
until completely dissolved, and stored at 4˚C. FTC was administered once-per-day subcutaneously at 30 mg/kg during the first month of ART (weeks 6–10 post-infection) and at 20 mg/kg
once per-day for the remainder of ART. Raltegravir (Isentress, provided by Merck & Co.) was
given orally at 250 mg/animal twice daily for the first month of ART, and at 150 mg/animal
twice daily for the remainder of ART.

Trained animal technicians administered all ART drugs, and veterinary staff closely monitored animals for adverse side effects, which were treated immediately at their discretion.

A few animals experienced elevated creatinine levels due to prolonged treatment with
PMPA, (A16144 and A16145 in the MAG + LT group, A16149 in the mock vaccine group, and
A16237 in the MAG + AC group) so these animals were promptly switched to tenofovir disoproxil fumarate (TDF, Fisher Scientific, catalog number: AC-5262), a prodrug of tenofovir that
is metabolized to PMPA.

TDF was resuspended in a solution of 15% Kleptose in water, at a concentration of 10.2
mg/mL, and pH adjusted to 4.1–4.3. The solution was then filter purified and stored at 4˚C or
frozen at -20˚C for long-term storage. TDF was administered once per-day subcutaneously at
5.2 mg/kg for the duration of ART.

## DNA vaccinations

**Particle-mediated epidermal delivery (PMED, or gene gun).** Vaccine and adjuvant plasmids were formulated onto gold particles as previously described and administered using the PowderJect® XR1 gene delivery device (PowderJect Vaccines, Inc.) [15]. Fur was shaved off of vaccination sites, which were then swabbed with alcohol prior to vaccine administration. Macaques were vaccinated over 16 epidermal sites along the lower abdomen and over the inguinal lymph nodes. Each animal received 32 μg of the MAG or Gag DNA vaccine co-formulated with 3.2 μg of plasmid expressing the LT adjuvant (2 μg MAG or Gag DNA + 0.2 μg LT per site).

**Intradermal electroporation (ID EP).** MAG, Gag, and adjuvant plasmids (rhIL-12, LTA1, expressed on one plasmid each, and hRALDH2/rhIL-33 and rhPD-1/rhCD80 co-expressed on one plasmid each) were prepared in a citrate buffer and administered via intradermal injection into the dermis above the quadriceps muscle on each leg. For the first vaccination, each macaque received 900 μg of the MAG or Gag DNA vaccine co-formulated with 900 μg of DNA expressing hRALDH2/rhIL-33, 900 μg of DNA expressing rhIL-12, and 162 μg of DNA expressing LTA1, evenly distributed over 3 injection sites per leg (300 μg MAG or Gag + 300 μg hRALDH2/rhIL-33 + 300 μg rhIL-12 + 54 μg LTA1 per site). For each subsequent vaccination, each macaque received 900 μg of the MAG or Gag DNA vaccine co-formulated with 900 μg of DNA expressing hRALDH2/rhIL-33, 900 μg of DNA expressing rhIL-12, 975 μg of DNA expressing rhPD-1/rhCD80 and 162 μg of DNA expressing LTA1, evenly distributed over 4 injection sites per leg (225 μg MAG or Gag + 225 μg hRALDH2/rhIL-33 + 225 μg rhIL-12 + 244 μg rhPD-1/CD80 + 40.5 μg LTA1 per site). Prior to each vaccination, fur covering the vaccination site was shaved and the skin was swabbed with alcohol. Following injection of vaccine and adjuvant DNA, electrical pulses were delivered using the Agile Pulse device (BTX, catalog number: 47-0400N) according to the device manufacturer's instructions.

## Enzyme-linked immunospot assay (ELISpot)

ELISpot was performed to quantify the frequency of SIV-specific IFN-γ spot-forming cells (SFC) in accordance with previously described methods. In brief, PBMCs were isolated from whole blood via density gradient separation and stimulated with pools of 15-mer peptides overlapping by 11 amino acids and corresponding to the following SIVmac239 proteins: Gag, Env, Pol, Vif, Vpr, Rev, Nef, and Tat (NIH AIDS Reagent Program, catalog numbers as follows: Gag: 6204, Env: 6883, Pol: 6443, Vif: 6205, Vpr: 6449, Rev: 6448, Nef: 8762, Tat: 6207). As a negative control, samples were stimulated with dimethyl sulfoxide (DMSO). For a positive control, samples were stimulated with concanavalin A (5 μg/mL, Sigma-Aldrich, catalog number: C2272). Samples were considered positive if peptide-specific responses were at least twice that of the negative control plus at least 0.01% after background (DMSO) subtraction.

## Enzyme-linked immunosorbent assay (ELISA) for analysis of antibody responses and microbial translocation

SIV Env-specific IgG binding antibody was measured by ELISA, as previously described [15]. In brief, 1μg/mL SIVmac239 gp130 (NIH AIDS Reagent Program, catalog number: 12797) was used as the capture antigen, and a rabbit anti-IgG (heavy and light chains conjugated to horseradish peroxidase) was used to detect antibody bound to the capture antigen.

## Intracellular cytokine staining (ICS)

Cryopreserved PBMCs and MLN lymphocytes were thawed and rested at 37°C and 5% $CO_2$ for 6 hours before stimulation with DMSO, PMA/Ionomycin, or SIVmac239 Gag or Env peptides (1 μg/mL) for 1 hour with CD107a PECy5 (eBioH4A3, Thermofisher, catalog number: 15-1079-42) in R10 media before adding 1 mg/mL of Brefeldin A (Sigma-Aldrich®, catalog number: B7651-5MG). Cells were stimulated overnight (approximately 14 hours) at 37°C and 5% $CO_2$. After stimulation, cells were washed with PBS and stained using LIVE/DEAD® Aqua (Life Technologies, catalog number: L34957) amine-reactive dye to distinguish live cells, then surface stained with CD3 Brilliant Violet (BV) 711 (Sp34-2, BD Biosciences, catalog number: 740807), CD4 PerCPCy5.5 (L200, BD Biosciences, catalog number: 552838), CD8 APC-Cy7 (RPA-T8, BD Biosciences catalog number: 557760), CD28 PE-CF594 (CD28.2, BD Biosciences, catalog number: 562296), CD95 BV421 (Dx2, BD Biosciences, catalog number: 562616), in Brilliant Stain buffer (BD Biosciences, catalog number: 566349). Cells were then permeabilized with Cytofix/Cytoperm (BD Biosciences, catalog number: 554722) and stained for intracellular cytokines with an antibody cocktail of IFNγ FITC (B27, BD Biosciences, catalog number: 552887), TNFα PE-Cy8 (Mab11, BD Biosciences, 557647), IL-2 PE (MQ1-17H12, BD Biosciences, catalog number: 500307), and GranzymeB APC (GB12, ThermoFisher®, catalog number: MHGb05), in Perm/Wash™ Buffer (BD Biosciences, catalog number: 554723). Finally, cells were washed with Perm/Wash™ Buffer and fixed with 1% paraformaldehyde. Data were collected on an LSR II (BD Biosciences) and analyzed using FlowJo software (Version 9.7.6, Treestar Inc.).

## Intracellular cytokine staining (ICS) of gut mucosa

Intraepithelial and lamina propria lymphocytes were isolated from colon biopsies and stimulated in the presence of brefeldin A (Sigma-Aldrich®, catalog number: B7651-5MG) and CD107a antibody (eBioH4A3; eBioscience, catalog number: 15-1079-42), as previously described [72]. Cells were assessed for viability using LIVE/DEAD® Aqua (Life Technologies, catalog number: L34957) and stained using surface and intracellular/intranuclear markers as previously described [72]. All samples were acquired on an LSRII (BD Biosciences) and analyzed using FlowJo software version 9.9.4 (FlowJo; LLC). Gating schemes are described previously [72]. Briefly, CD4+ Tregs were designated by coexpression of CD25 and FoxP3 and Th17 cells were defined by IL-17 production.

## Statistical analyses

Statistical differences between multiple groups were calculated using a Dunn's multiple comparisons test, while statistical comparisons between two groups were determined using a two-sided Mann-Whitney. Statistical differences in viral load, CD4+ T cell counts, or immune responses between time points were calculated using a Wilcoxon matched-pairs signed-rank test. Viral burden was determined by calculating the area under the curve of each animal's viral load graph. Correlations between immune responses and viral burden were determined by a Spearman's rank correlation test. When necessary, P values were adjusted for multiple comparisons using the Benjamini-Hochberg method. A P value of $\leq 0.05$ was considered significant for each test. All calculations were performed using GraphPad Prism software (Version 8, GraphPad Software).

## Supporting information

**S1 Fig. Animals in each vaccine group demonstrate similar plasma viral loads, CD4+ T cell counts, and ART responsiveness. (A)** Plasma viral loads were determined by RT-q-PCR for

the mock (black circles), MAG + LT (red squares) and MAG + AC (purple triangles) groups. The dashed line indicates the assay limit of detection (30 viral RNA copies/1mL of plasma) and the dotted line indicates the threshold for control of virus replication. Shown is the decrease of each animals' viral loads between pre-ART (6 wpi) and pre-vaccination (32 wpi). Statistical analyses were performed using a Wilcoxon matched-pairs signed rank test; results are considered significant if $P \leq 0.05$. **(B)** Percent of baseline CD4+ T cell counts were calculated for the mock, MAG + LT and MAG + AC groups over time by dividing the absolute CD4 + count at a timepoint by the absolute CD4+ count at 0 wpi and multiplying by 100. The dotted line indicates 50% of baseline CD4+ T cells. CD4 T cell counts were obtained using a Beckman Coulter® AC*T™ 5diff hematology analyzer. Shown is the restoration of each animals' percent of baseline CD4+ T cell counts between pre-ART (6 wpi) and pre-vaccination (32 wpi). Statistical analyses were performed using a Wilcoxon matched-pairs signed rank test; results are considered significant if $P \leq 0.05$.
(TIF)

**S2 Fig. The MAG + AC group shows a trend towards increased IFN-γ and TNFα CD8+ T cell responses post-vaccination compared to pre-vaccination. (A)** PBMCs were thawed and stimulated with Gag peptides, and expression of IL-2, IFN-γ, TNFα and CD107a/GzB were quantified using intracellular cytokine staining. Shown are the medians and interquartile ranges of each group's SIV-Gag specific T cell response. **(B)** Lymphocytes isolated from MLN were thawed and stimulated with Gag peptides, and expression of IL-2, IFN-γ, TNFα and CD107a/GzB were quantified using intracellular cytokine staining. Shown are the medians and interquartile ranges of each group's SIV-Gag specific T cell response. **(A, B) Statistics**. Statistical comparisons between baseline and post-vaccination timepoints within a group were calculated using a Wilcoxon matched-pairs signed rank test. Results are considered significant if $P \leq 0.05$.
(TIF)

**S3 Fig. No differences were observed between groups in Gag-specific CD4+ T cell responses in PBMC and MLN. (A)** PBMCs were thawed and stimulated with Gag peptides, and expression of IL-2, IFN-γ, TNFα and CD107a/GzB were quantified using intracellular cytokine staining. Shown are the medians and interquartile ranges of each group's SIV-Gag specific T cell response. **(B)** Lymphocytes isolated from MLN were thawed and stimulated with Gag peptides, and expression of IL-2, IFN-γ, TNFα and CD107a/GzB were quantified using intracellular cytokine staining. Shown are the medians and interquartile ranges of each group's SIV-Gag specific T cell response. **(A, B) Statistics**. Statistical comparisons between baseline and post-vaccination timepoints within a group were calculated using a Wilcoxon matched-pairs signed rank test. A Dunn's multiple comparisons test was used when making multiple comparisons between vaccine groups and the mock group. Results are considered significant if $P \leq 0.05$.
(TIF)

**S4 Fig. No differences were observed among groups in Env-specific CD8+ or CD4+ T cell responses in PBMC. (A-B)** PBMCs were thawed and stimulated with Env peptides, and expression of IL-2, IFN-γ, TNFα and CD107a/GzB were quantified using intracellular cytokine staining. Shown are the medians and interquartile ranges of each group's SIV-Env specific T cell response. **(A, B) Statistics**. Statistical comparisons between baseline and post-vaccination timepoints within a group were calculated using a Wilcoxon matched-pairs signed rank test. A Dunn's multiple comparisons test was used when making multiple comparisons between vaccine groups and the mock group. Results are considered significant if $P \leq 0.05$.
(TIF)

**S5 Fig. No differences were observed among groups in Env-specific CD8$^+$ or CD4$^+$ T cell responses in MLN. (A-B)** Lymphocytes isolated from MLNs were thawed and stimulated with Env peptides, and expression of IL-2, IFN-γ, TNFα and CD107a/GzB were quantified using intracellular cytokine staining. Shown are the medians and interquartile ranges of each group's SIV-Env specific T cell response. **(A, B) Statistics**. Statistical comparisons between baseline and post-vaccination timepoints within a group were calculated using a Wilcoxon matched-pairs signed rank test. A Dunn's multiple comparisons test was used when making multiple comparisons between vaccine groups and the mock group. Results are considered significant if P $\leq$ 0.05.
(TIF)

**S6 Fig. The MAG + AC group shows a trend towards increased IFN-γ T cell responses post-vaccination as measured by ELISpot and ICS. (A)** PBMCs were thawed and stimulated with Gag peptides and expression of IFN-γ was quantified using intracellular cytokine staining. Shown are the medians and interquartile ranges of each group's SIV-Gag specific T cell response. **(B)** Bulk PBMCs were stimulated with Gag peptides to quantify the SIV-specific IFN-γ response. Results were considered positive if peptide-specific responses were at least twice that of the negative control plus at least 0.01% after background (DMSO) subtraction. Shown are medians and interquartile ranges with data from individual animals layered over each bar. **(A-B)** A Dunn's multiple comparisons test was used when making multiple comparisons between vaccine groups and the mock group. Results are considered significant if P $\leq$ 0.05.
(TIF)

**S7 Fig. The MAG + AC group shows a trend towards increased SIV-specific polyfunctional CD8$^+$ T cell responses post-vaccination in PBMC and MLN. (A-B)** PBMCs and lymphocytes from MLNs were thawed and stimulated with Gag and Env peptides, and expression of IL-2, IFN-γ, TNFα and CD107a/GzB were quantified using intracellular cytokine staining. Polyfunctionality is defined as the frequency of T cells specific for Gag or Env and expressing any three or more effector functions. Shown are medians and interquartile ranges with data from individual animals layered over each bar. A Dunn's multiple comparisons test was used when making comparisons between vaccine groups and the mock group. Results are considered significant if P $\leq$ 0.05.
(TIF)

**S8 Fig. CD4$^+$ T cell counts corresponded with virus burden in plasma. (A-C)** Shown are the percent of baseline CD4$^+$ T cell counts for each individual animal in the mock, MAG + LT and MAG + AC groups over time. Percent of baseline CD4$^+$ T cell counts were calculated for the mock, MAG + LT and MAG + AC groups over time by dividing the absolute CD4$^+$ count at a timepoint by the absolute CD4$^+$ count at 0 wpi and multiplying by 100. The dotted line indicates 50% of baseline CD4$^+$ T cells. CD4$^+$ T cell counts were obtained using a Beckman Coulter$^®$ AC*T™ 5diff hematology analyzer. **(D)** Graphed are the median and interquartile range of controllers' and non-controllers' percent of baseline CD4$^+$ counts.
(TIF)

**S9 Fig. Excluding animals with incomplete viral suppression on ART does not change our conclusions on vaccine efficacy. (A-C)** Plasma viral RNA was quantified using RT-q-PCR, with a limit of detection of 30 viral RNA copies per 1 mL of plasma, as indicated by the dashed line. ART low responders A16234 and A16236 were removed from the mock-vaccinated group and the MAG + AC group, respectively. **(D)** Shown are the median viral load and interquartile ranges for each treatment group. The dotted line indicates the threshold for control of

virus replication, based on previous studies using SIVΔB670.
(TIF)

**S10 Fig. Non-controllers exhibited a trend towards higher median titers of Env-specific IgG compared to controllers.** The magnitude of the SIV Env-specific IgG response in the plasma was measured by ELISA, using SIV gp130 as the capture antigen. Shown are medians and interquartile ranges.
(TIF)

**S11 Fig. Controllers and non-controllers demonstrate similar levels of SIV Gag-specific CD4$^+$ T cell responses in the PBMC and MLN post-vaccination and during ATI. (A-B)** PBMCs and MLNs were thawed and stimulated with Gag peptides, and expression of cytokines was quantified using intracellular cytokine staining. Shown are the medians and interquartile ranges of the SIV Gag-specific CD4$^+$ T cell responses of controllers and non-controllers, with individual responses layered over each pre-ATI (50 wpi) and during ATI (62 wpi for PBMC and 66 wpi for MLN). Statistical differences between controllers and non-controllers at each timepoint were assessed using a Mann Whitney t test and the Benjamini-Hochberg method was used to adjust P values. Results are considered significant if P $\leq$ 0.05.
(TIF)

**S12 Fig. No difference between controllers/non-controllers in Gag-specific IL-2$^+$ and CD107a$^+$GzB$^+$ CD8$^+$ T cells in PBMC or Gag-specific IFN-$\gamma^+$, TNF$\alpha^+$, and CD107a$^+$GzB$^+$ CD8$^+$ T cells in MLN. (A-E)** PBMCs and MLNs were thawed and stimulated with Gag peptides, expression of cytokines was quantified using intracellular cytokine staining. Shown are the medians and interquartile ranges of the SIV Gag-specific CD8$^+$ T cell responses of controllers and non-controllers, with individual responses layered over each bar pre-ATI (50 wpi) and during ATI (62 wpi for PBMC and 66 wpi for MLN). Statistical differences between controllers and non-controllers at each timepoint were assessed using a Mann Whitney t test and the Benjamini-Hochberg method was used to adjust P values. Results are considered significant if P $\leq$ 0.05.
(TIF)

**S13 Fig. Non-polyfunctional responses compose the majority of the SIV-specific CD8$^+$ T cell response but are not elevated in controllers compared to non-controllers. (A-D)** PBMCs and lymphocytes from MLNs were thawed and stimulated with Gag and Env peptides, and intracellular cytokine staining was used to quantify expression of IL-2, IFN-$\gamma$, TNF$\alpha$ and CD107a/GzB. SIV-specific T cells expressing three or more effector functions are considered polyfunctional, while SIV-specific T cells expressing two or fewer effector functions are considered non-polyfunctional. Shown are medians and interquartile ranges with data from individual animals layered over each bar. At each timepoint, differences between controllers and non-controllers were assessed using a Mann Whitney t test, and the Benjamini-Hochberg method was used to adjust P values. Results are considered significant if P $\leq$ 0.05.
(TIF)

**S14 Fig. Baseline frequencies of Th17 and Treg cells are not associated with therapeutic outcome during ATI. (A-B)** Lymphocytes were isolated from colon biopsies and expression of Th17 and Treg markers was quantified using intracellular cytokine staining on fresh cells. A Spearman rank correlation test was used to determine P and r values. Shown are the Benjamini-Hochberg adjusted P values, results are considered significant if P $\leq$ 0.05.
(TIF)

**S1 Table. MHC and TRIM5 genetics.** Indicated for each animal are TRIM5 haplotype, the presence or absence of an MHC-1 allele commonly associated with control of viral replication, vaccination group, and virological status during ATI. Animals in gray are non-controllers, while controllers are shown in green.
(XLSX)

## Acknowledgments

The authors would like to thank all the veterinary and research support staff of the Washington National Primate Research Center (WaNPRC), with special thanks given to Solomon Wangari, Drew May, Dr. Jennifer Lane, Dr. Cassie Moats, Dr. Jeremy Smedley, and Dr. Robert Murnane. We thank Thomas Lewis and Dr. Patience Murapa for their early work in determining the optimal adjuvant combination for use in this study. We also thank the NIH AIDS Research and Reference Reagent Program for providing all the SIV peptides and SIV gp130 proteins used in this study, free of charge, and Gilead Sciences and Merck & Co. for their gracious donations of ART drugs.

## Author Contributions

**Conceptualization:** Hillary Claire Tunggal, Kenneth Bagley, Deborah Heydenburg Fuller.

**Data curation:** Hillary Claire Tunggal, Paul Veness Munson, Megan Ashley O'Connor, Deborah Heydenburg Fuller.

**Formal analysis:** Hillary Claire Tunggal, Paul Veness Munson, Megan Ashley O'Connor.

**Funding acquisition:** Megan Ashley O'Connor, Kenneth Bagley, Deborah Heydenburg Fuller.

**Investigation:** Hillary Claire Tunggal, Paul Veness Munson, Megan Ashley O'Connor, Nika Hajari.

**Methodology:** Hillary Claire Tunggal, Paul Veness Munson, Megan Ashley O'Connor, Sandra Elizabeth Dross, James Thomas Fuller.

**Project administration:** Debra Bratt, Deborah Heydenburg Fuller.

**Resources:** Debra Bratt, James Thomas Fuller.

**Supervision:** Deborah Heydenburg Fuller.

**Validation:** Hillary Claire Tunggal, Paul Veness Munson.

**Visualization:** Hillary Claire Tunggal, Paul Veness Munson.

**Writing – original draft:** Hillary Claire Tunggal.

**Writing – review & editing:** Hillary Claire Tunggal, Paul Veness Munson, Megan Ashley O'Connor, Kenneth Bagley, Deborah Heydenburg Fuller.

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
