## [Decision Letter · Decision Letter 0]

2 Mar 2021

PONE-D-21-01499

Effects of therapeutic vaccination on the control of SIV in rhesus macaques with variable responsiveness to antiretroviral drugs.

PLOS ONE

Dear Dr. Fuller,

Thank you for submitting your manuscript to PLOS ONE. After careful consideration, we feel that it has merit but does not fully meet PLOS ONE’s publication criteria as it currently stands. Therefore, we invite you to submit a revised version of the manuscript that addresses the points raised during the review process.

Both reviewers found your manuscript is interesting but raised several questions (see below), and especially some of the reviewers suggest to focus on the immunogenicity of the multi antigen vaccine and discuss reasons behind this novel combination of adjuvants did not work in the setting of a therapeutic vaccine?

We look forward to receiving your revised manuscript.

Kind regards,

Siddappa N. Byrareddy, PhD

Academic Editor

PLOS ONE

Journal Requirements:

2. In order to comply with PLOS ONE's guidelines for non-human primate experiments (http://journals.plos.org/plosone/s/submission-guidelines#loc-non-human-primates), we kindly request that you provide specific details regarding housing conditions (e.g. cage sizes, whether housed alone), feeding regimens and the specific environmental enrichment provided. In addition, please specify the disposition of animals at the end of the study (e.g. euthanasia, returned to home colony, etc.). If animals were euthanized following the study, please provide the method of sacrifice.

3.Thank you for stating the following in the Competing Interests section:

"Dr. Kenneth Bagley was a paid employee of Profectus Biosciences and had stock options with Profectus when the research was performed. This study was supported by an SBIR grant awarded to Profectus, of which Dr. Bagley was the Principal Investigator. Dr. Bagley's employment with Profectus was terminated in August 2019 and he did not exercise any stock options. "

We note that one or more of the authors are employed by a commercial company: Orlance Incorporation and Profectus Biosciences

Reviewers' comments:

Reviewer's Responses to Questions

**Comments to the Author**

1. Is the manuscript technically sound, and do the data support the conclusions?

Reviewer #1: Partly

Reviewer #2: No

2. Has the statistical analysis been performed appropriately and rigorously? 

Reviewer #1: Yes

Reviewer #2: Yes

3. Have the authors made all data underlying the findings in their manuscript fully available?

Reviewer #1: Yes

Reviewer #2: Yes

4. Is the manuscript presented in an intelligible fashion and written in standard English?

Reviewer #1: Yes

Reviewer #2: Yes

5. Review Comments to the Author

Reviewer #1: This manuscript describes the therapeutic impact of DNA vaccinations on SIV-infected rhesus macaques. In this study, 14 rhesus macaques were infected with SIV∆670 and started on ART 6 weeks post-infection. Most animals maintained low but detectable viral loads during ART. At 50 weeks post-infection, the animals were divided into three groups: (1) DNA vaccination adjuvanted with E. coli enterotoxin, (2) DNA vaccination with a combination of adjuvants, or (3) mock vaccinated. The animals received five monthly DNA immunizations encoding SIV Gag-Pol-Env, with the adjuvant combination designed to maximize mucosal and systemic T cell responses. Vaccination induced transient increases in peripheral SIV-specific T cell responses, peaking before the final immunization. Although the immunizations may have skewed T cell responses toward Gag and Env. However, after stopping ART, the vaccine-induced immune responses failed to prevent viral rebound or significantly reduce virus replication. The authors performed more in-depth analyses of macaques controlling SIV post-ART, regardless of vaccination status. Lower viral burdens and poly-functional CD8+ T cells pre-ART and Th17/Treg CD4+ T cell ratio in the colon pre-infection were associated with post-ART virus suppression.

Overall, the authors provide a thorough analysis of cellular immunity during therapeutic vaccinations of SIV-infected rhesus macaques. However, it is challenging to differentiate vaccine-induced responses from those generated by persistent virus replication. Details about the study design are missing; clarifying these points will strengthen the manuscript. Small experimental groups make it challenging to contextualize the results in broader HIV cure programs. Some comparisons were statistically significant but skewed by a single animal within a group, making it difficult to draw firm conclusions from the results. Therefore, the authors should temper some of their conclusions.

1) The source of the vaccine immunogens is not mentioned in the text. Are they derived from SIV∆670 or another SIV strain? Did the peptides used in the ELISPOT/ICS assays match these immunogens and/or SIV∆670? If not, how may have heterologous antigens/peptides impact the results?

2) What is the rationale for delivering Gag p57 separately?

3) SIV∆670 is infrequently used in macaque studies. How susceptible is this strain to ARVs? Did the inherent ARV resistance of SIV∆670 contribute to persistent viral loads during ART? The susceptibility of SIV∆670 should be discussed in the text.

4) The figure legend for Figure 2 describes panels B and C as containing the cumulative IFNg responses after peptide stimulation, both displaying week 50. Why are the Y axes different? Is the animal with ~6000 SFC in figure 2C missing from 2B? Is the animal with 10^4 viral loads one with high SFC counts in 2C? If so, is it possible to distinguish vaccine-induced responses from those stimulated by ongoing virus replication?

5) The text in line 287 states that post-ATI viral control is primarily mediated by CD8+ T cells. This comment should be tempered as only bulk Env-binding antibodies were analyzed. It is formally possible that strain-specific neutralizing or non-neutralizing antibodies contributed to viral suppression.

6) Persistent virus replication could have increased viral diversity within non-controllers relative to controllers, potentially leading to immune escape and an inability to control virus replication post-ATI. This possibility should be addressed in the discussion section.

7) In line 390, it is more appropriate to refer to these animals as controlling virus replication post-ATI than being protected from rebound since they had detectable viremia.

Minor comments:

Line 239: should “that” replace “this” in the phrase “an outcome this is likely due to the…”?

Line 268: references supplemental figure 7, but the data appears to be in supplemental figure 9

Line 277: references supplemental figure 6, but the data appears to be in supplemental figure 8

Reviewer #2: The manuscript by Tunggal, et al., describes the preclinical evaluation of a therapeutic DNA-based HIV vaccine in SIV-infected rhesus macaques (RM) on antiretroviral therapy (ART). They evaluated two novel DNA vaccine constructs, the first containing plasmids expressing SIV proteins Gag, Pol and Env (MAG) adjuvanted with a plasmid expressing the catalytic subunit of enterotoxin (LTA1), referred to as MAG + LT. The second construct consisted of SIV DNA plasmid (MAG) adjuvanted with plasmids encoding (LTA1), the cytokines IL-12 and IL-33, the enzyme 85 retinaldehyde dehydrogenase 2 (RALDH2), soluble PD-1 (sPD-1), and soluble CD80 (sCD80) referred to as MAG + AC. Both vaccines were administered in 5 monthly doses starting 32 weeks post-infection and the impact of this therapeutic vaccination regimen on protection from virus rebound following ART interruption was assessed. They found no significant difference in protection from rebound or levels of post-ART viremia between vaccinated vs. control RM. However, due to the variability in post-ART viral loads, they examined immune correlates of viral control and showed higher frequencies of colonic CD4+ T cells and lower Th17/Treg ratios prior to infection correlated with improved post-ART viral control. While the study provides some intriguing observations, there are a few concerns.

Major concerns

1. The 2 animals with incomplete viral suppression (mean pvl > 10e4) should be excluded from all analyses. If the criteria for determining therapeutic vaccine efficacy was containment of pvl < 10e3 copies/ml during ATI, then why include animals in the analysis that did not meet this criterion prior to ART withdrawal.

2. Along those lines, the IFN-g ELISPot data showed a significant difference in responses between MAG + AC and MAG + LT vaccine groups at 50 wpi (Fig 2B-C). However, the ICS data showed no difference between vaccine groups at any time point (S2-5 Figs). The reason for this discrepancy is unclear and was not discussed? Also, why were SIV Pol responses not readily detectable in either vaccine group considering Pol was included in the vaccine construct?

3. The comparison of CD8+ T cell polyfunctional responses was unclear. What proportion of CD8+ T cell responses in controllers were not polyfunctional? Also, did the vaccine improve polyfunctionality of the CD8+ T cell response? A treatment group comparison of CD8+ T cell polyfunctional responses should have been presented to bolster the correlational analysis and resultant conclusions.

4. Again, it is difficult to make the case that mucosal immune responses prior to ART suppression influence virologic outcome during and post-ART (Fig 7) if the immune corelate analysis includes animals with ongoing viremia during ART. Is it unclear how the frequencies of colonic CD4+ T cells or lower Th17/Treg ratios prior to infection would affect the ability of ART to suppress viremia. By including the 2 animals with incomplete virus suppression, it is difficult to reach any of the conclusions outlined.

5. Another major issue is the fact that the controllers had lower viral loads at the time of ART initiation (Fig 4). This would suggest that these animals already had effective T cell responses prior ART initiation, which were maintained during therapy and helped facilitate better post-ART control. So, the suggestion that “pre-infection immune parameters helped animals develop polyfunctional CD8+ T cell responses during ART” (lines 339 – 342) is not supported by the data presented.

Minor concerns

• The manuscript should be checked for grammatical errors.

• Line 96 needs clarification as is says PD-1 and CTLA4 can “restore” immune exhaustion.

• Line 268 should be “S9 Fig.” not “S7 Fig.”.

• Line 277 says “S6 Fig.” but the figure which is supposed to show the correlation between viral control and Gag-specific CD4+ T cell responses is missing.

• There was no data or discussion on the impact of therapeutic vaccination on cell-associated viral loads. Did the controllers have a lower reservoir size? If so, perhaps this could also contribute to improved post-ART viral control.

6. PLOS authors have the option to publish the peer review history of their article (what does this mean?). If published, this will include your full peer review and any attached files.

Reviewer #1: No

Reviewer #2: No

---

## [Author Response · Author response to Decision Letter 0]

6 Apr 2021

We thank the reviewers for their insightful and constructive critique of this manuscript. To address each concern, we edited the text as indicated below, modified our figures and discussion, and added new data as needed. We believe these revisions fully address the reviewers’ concerns and improve the focus and impact of the manuscript.

Response: We have edited the manuscript to meet PLOS ONE’s style requirements.

2. In order to comply with PLOS ONE's guidelines for non-human primate experiments (http://journals.plos.org/plosone/s/submission-guidelines#loc-non-human-primates), we kindly request that you provide specific details regarding housing conditions (e.g. cage sizes, whether housed alone), feeding regimens and the specific environmental enrichment provided. In addition, please specify the disposition of animals at the end of the study (e.g. euthanasia, returned to home colony, etc.). If animals were euthanized following the study, please provide the method of sacrifice.

Response: We revised the Ethics Statement to include the requested information, including the number of animals, feeding regimen, specific housing conditions and environmental enrichments, humane endpoint criteria, and method of euthanasia. We also combined the Ethics Statement with the section on viral challenge and AIDS monitoring in order to make the methods section clearer and more concise. 

The following text was added and edited (Materials and methods, Pages 24-25, Lines 548-570):

Animals were singly housed in stainless-steel cages with a minimum of 6.0sq ft of floor space. In animal housing areas, humidity was maintained between 30 - 70% while temperature was kept at a range of 72 - 82�F. Paired animals were kept in adjacent cages to allow for grooming contact. Waste pans were cleaned daily, while cages, racks, and accessories were sanitized in mechanical cage washers at least once every two weeks. Animals were fed a commercial monkey chow, supplemented daily with fruits and vegetables, and drinking water was available at all times. Environmental enrichment was provided throughout the duration of the study, including grooming contact, perches, toys, foraging experiences, and food enrichment. Animal care staff monitored the health and well-being of all animals on a daily basis. All biopsies, surgeries, and blood draws were performed under ketamine anesthesia (10 mg/kg) and any continuous discomfort or pain was alleviated at the discretion of the veterinary staff to minimize suffering. 

Rhesus macaques were challenged intravenously with 100 TCID50 of cryopreserved SIV�B670, diluted in 1 milliliter of RPMI. Following SIV infection, animals were monitored for changes in weight, complete blood counts, blood CD4+ T cell count, and clinical signs of opportunistic infections. Simian AIDS was defined according to WaNPRC guidelines, namely: weight loss exceeding 15%, anemia (sustained hematocrit <15%), CD4+ T cell decline to less than 200 cells per microliter, and presence of opportunistic infections. All four criteria were evaluated in each animal at least monthly and at each specimen collection time point, or more frequently if two or more signs of SIV disease progression were present. None of the animals required euthanasia prior to their experimental endpoint at 80 weeks post-infection.

At the experimental endpoint, euthanasia was performed by administration of Euthasol� (Virbac Corp., Houston, TX) while the animal was under deep anesthesia, in accordance with the 2007 American Veterinary Medical Association Guidelines on Euthanasia.

3.Thank you for stating the following in the Competing Interests section:

"Dr. Kenneth Bagley was a paid employee of Profectus Biosciences and had stock options with Profectus when the research was performed. This study was supported by an SBIR grant awarded to Profectus, of which Dr. Bagley was the Principal Investigator. Dr. Bagley's employment with Profectus was terminated in August 2019 and he did not exercise any stock options."

We note that one or more of the authors are employed by a commercial company: Orlance Incorporation and Profectus Biosciences.

“The funder provided support in the form of salaries for authors [insert relevant initials] but did not have any additional role in the study design, data collection and analysis, decision to publish, or preparation of the manuscript. The specific roles of these authors are articulated in the ‘author contributions’ section.”

Response: We appreciate the opportunity to clarify our commercial affiliations. Dr. Kenneth Bagley was involved in the study design and preparation of this manuscript while employed at Profectus Biosciences. Dr. Bagley is now employed at Orlance Incorporated, which was cofounded by Dr. Deborah Fuller. However, Orlance did not provide any support for this study and did not play any role in the study design, data collection or analysis, decision to publish, or preparation of the manuscript. We amended our Funding Statement below to more clearly state and explain the roles of our commercial affiliations in this study.

Funding Statement: This work was supported with federal funds from the National Institutes of Health (www.nih.gov, T32-AI007140 to M.A.O.), the National Institute of Allergy and Infectious Diseases (www.niaid.nih.gov, R44AI110315 to K.B. and R01 AI104679 to D.H.F.), and the Office of Research Infrastructure Programs (orip.nih.gov, P51OD010425 to D.H.F.). During this study, K. B. was a paid employee of Profectus Biosciences and assisted in study design and participated in the review and editing of this manuscript. Apart from these specific roles, Profectus Biosciences did not have any additional role in the data collection, data analysis or decision to publish the manuscript. K.B. is currently a paid employee of Orlance Incorporation. Orlance did not provide any support for this study and was not involved in the study design, data collection and analysis, decision to publish, or preparation of the manuscript. The other funders of this study provided support in the form of salaries for authors H.C.T., P.V.M., M.A.O., N.H., S.E.D., J.T.F., D.B., and D.H.F., but did not have any additional role in the study design, data collection and analysis, decision to publish, or preparation of the manuscript. The specific roles of these authors are articulated in the ‘author contributions’ section. 

Please know it is PLOS ONE policy for corresponding authors to declare, on behalf of all authors, all potential competing interests for the purposes of transparency. PLOS defines a competing interest as anything that interferes with, or could reasonably be perceived as interfering with, the full and objective presentation, peer review, editorial decision-making, or publication of research or non-research articles submitted to one of the journals. Competing interests can be financial or non-financial, professional, or personal. Competing interests can arise in relationship to an organization or another person. Please follow this link to our website for more details on competing interests:

http://journals.plos.org/plosone/s/competing-interests

Response: We included an updated Competing Interests Statement below, declaring commercial affiliations with Profectus Biosciences and Orlance Incorporated and confirming that these commercial affiliations do not alter our adherence to all of PLOS ONE’s policies on sharing data and materials. 

Competing Interests Statement: Dr. Kenneth Bagley was a paid employee of Profectus Biosciences and had stock options with Profectus when the research was performed. A portion of this study was supported by the SBIR R44AI110315 awarded to Profectus BioSciences. Dr. Bagley was the Principal Investigator for R44AI110315. At the time the study was performed, Dr. Bagley had unexercised stock options with with Profectus BioSciences. Dr. Bagley left Profectus BioSciences in August 2019. He did not exercise any stock options and no longer has an interest in Profectus BioSciences. Dr. Kenneth Bagley is currently a paid employee of Orlance Incorporated, which was co-founded by D.H.F. This study was not supported in any way by Orlance. The affiliations of Dr. Bagley with Profectus and Orlance and Dr. Fuller’s co-founder interests in Orlance do not alter our adherence to PLOS ONE policies on sharing data and materials. There are no patent applications (pending or actual) affiliated with this study.

Reviewer 1 Comments:

1) The source of the vaccine immunogens is not mentioned in the text. Are they derived from SIV∆670 or another SIV strain? Did the peptides used in the ELISPOT/ICS assays match these immunogens and/or SIV∆670? If not, how may have heterologous antigens/peptides impacted the results?

Response: We amended the text to specify the source of the vaccine immunogens and the peptides used in the ELISpot/ICS assays and provided a brief discussion of how the heterogeneity between the SIV inoculum, vaccine source, and peptide source may have impacted the results. The following text was added or modified to address these points.

(Results, Page 6, Lines 120-124):

Rhesus macaques were intravenously infected with SIVΔB670, a highly pathogenic, primary isolate that induces AIDS in most rhesus macaques within 5-17 months of infection [38]. This strain was chosen because the 15% divergence between the Env consensus sequence of the SIVΔB670 inoculum and the SIV/17E-Fr Env sequence in the vaccine mimics therapeutic vaccination of humans infected with diverse variants of a given HIV subtype [15, 38, 39]. (Results, Page 10, Lines 217-224): 

To determine the impact of the therapeutic vaccines on T cell responses, SIV-specific T cells producing IFN-γ in response to stimulation with peptide pools spanning SIVmac239 p57Gag (Gag), gp130 (Env), Pol, Vif, Vpr, Rev, Nef, and Tat were measured by ELISpot. Although the sequences are not a precise match for the SIVΔB670 inoculum or the SIV/17E-Fr strain used to construct the DNA vaccine, the genetic diversity between the three viruses resembles the intraclade diversity observed for HIV-1 [38] and thus are relevant in the context of human HIV-1 infection. However, we cannot rule out the possibility that some vaccine or challenge strain-specific IFN-γ responses were not detected in our T cell assays due to these differences.

2) What is the rationale for delivering Gag p57 separately?

Response: The following text was added to explain our rationale (Results, Page 7, Lines 146-152): 

The MAG vaccine was supplemented with an additional plasmid encoding p57 Gag based on evidence from both human and NHP studies showing that Gag-specific T cell responses are crucial for control of viral replication [15, 41-45]. Additionally, our previously published therapeutic vaccine study demonstrated that SIVΔB670-infected rhesus macaques vaccinated with a Gag expression plasmid within the context of a multi-epitope SIV DNA vaccine durably controlled viral rebound after stopping ART and consistently demonstrated elevated Gag-specific CD8+ T cell responses [39].

3) SIV∆670 is infrequently used in macaque studies. How susceptible is this strain to ARVs? Did the inherent ARV resistance of SIV∆670 contribute to persistent viral loads during ART? The susceptibility of SIV∆670 should be discussed in the text.

Response: We thank the reviewer for raising this point since more recent studies in our lab indicate this was likely a factor that influenced our results. We revised the text to address the susceptibility of SIV∆670 to ART, including how the levels of suppression we achieved here compare to what was seen in prior studies, and in comparison to more commonly used SIVs. (Results, Page 8, Lines 178-188):

Notably, our previous studies indicate that SIVΔB670 appears to be more resistant to ART [15, 39] than other SIVs commonly used as inoculums in NHP studies, namely SIVmac251 [48, 49] and SIVmac239 [50, 51]. This inherent ART resistance likely contributed to persistent viral replication during ART in 2 out of 14 animals. However, the levels of viral suppression on ART in these animals were superior to our previous therapeutic studies utilizing SIVΔB670, including two studies where we reported a significant impact of therapeutic vaccination on viral control in ART responders [15, 39]. Our previous therapeutic vaccine studies with SIVΔB670 reported highly variable responses to ART consisting of one or two anti-retroviral drugs, with 40-50% of macaques exhibiting no decrease in viral load during ART [15, 39]. Here, using a more potent triple drug combination, 14/14 animals exhibited at least a 3-log decrease in viral load within 4 weeks of ART initiation that was maintained for the duration of ART.

4) The figure legend for Figure 2 describes panels B and C as containing the cumulative IFN-γ responses after peptide stimulation, both displaying week 50. Why are the Y axes different? Is the animal with ~6000 SFC in figure 2C missing from 2B? Is the animal with 104 viral loads one with high SFC counts in 2C? If so, is it possible to distinguish vaccine-induced responses from those stimulated by ongoing virus replication?

Response: We appreciate the reviewer pointing out these discrepancies between panels B and C in Figure 2 and corrected the figure to make the axes consistent between panels. The animal with ~6000 SFC in Figure 2C had been cut off when converting the figure into the proper format. We also added the following lines to address whether vaccine-induced responses were distinguishable from virus-induced responses (Results, Pages 10-11, Lines 231-236):

It was not possible to distinguish between virus-stimulated and vaccine-induced immune responses from these experiments. However, SIV-specific IFN-γ responses were mostly undetectable prior to vaccination (32 wpi), suggesting that most animals did not have these responses prior to vaccination. We thereby conclude that the increase in responses detected post-vaccination is due to the vaccine inducing de novo T cell responses or boosting virus-primed responses. 

5) The text in line 287 states that post-ATI viral control is primarily mediated by CD8+ T cells. This comment should be tempered as only bulk Env-binding antibodies were analyzed. It is formally possible that strain-specific neutralizing or non-neutralizing antibodies contributed to viral suppression.

Response: We thank the reviewer for raising this point. This statement made in the manuscript is based on the significant correlations we observed between increased CD8+ T cell responses measured right before stopping ART and lower viral burden post-ART. In contrast, we did not observe an association between binding antibody titers and viral control. Rather, we observed a trend towards higher binding antibody titers and increased viral burden. This trend is not statistically significant but suggests antibodies present prior to stopping ART likely had no impact on viral control post-ART. However, since strain-specific neutralizing and non-neutralizing antibody effector functions were not measured in this study, we cannot rule out a possible role for these responses in protection. We therefore amended the text in lines 287-288 to address this possibility. (Results, Pages 15-16, Lines 346-349):

Although we cannot completely rule out a role for strain-specific neutralizing or non-neutralizing antibodies, these results suggest viral control during ATI was likely mediated, at least in part, by CD8+ T cell responses.

6) Persistent virus replication could have increased viral diversity within non-controllers relative to controllers, potentially leading to immune escape and an inability to control virus replication post-ATI. This possibility should be addressed in the discussion section.

Response: We completely agree with this statement and thank the reviewer for raising this point. We revised the discussion to address this possibility. (Discussion, Page 22, Lines 495-501):

Although quantifying viral diversity was outside the scope of this study, persistent viral replication during ART likely increased viral diversity, especially in non-controllers relative to controllers. Higher viral diversity increases the likelihood of immune escape from vaccine-induced immune responses and may have contributed to failure to control viral replication during ATI in some animals. Additional studies are needed to determine how incomplete suppression of viral replication during ART impacted viral diversity in this study and its impact on therapeutic vaccine efficacy.

7) In line 390, it is more appropriate to refer to these animals as controlling virus replication post-ATI than being protected from rebound since they had detectable viremia.

This has been corrected.

Minor comments:

Line 239: Should “that” replace “this” in the phrase “an outcome this is likely due to the…”?

This has been corrected.

Line 268: References supplemental figure 7, but the data appears to be in supplemental figure 9.

This has been corrected.

Line 277: References supplemental figure 6, but the data appears to be in supplemental figure 8.

This has been corrected.

Reviewer 2 Comments:

Major concerns

1. The 2 animals with incomplete viral suppression (Mean PVL > 104) should be excluded from all analyses. If the criteria for determining therapeutic vaccine efficacy was containment of PVL < 103 copies/mL during ATI, then why include animals in the analysis that did not meet this criterion prior to ART withdrawal.

Response: We thank the reviewer for raising this point. In our previous therapeutic vaccine studies, we excluded animals with high levels of persistent viral replication from our analyses and observed a significant impact of therapeutic vaccines on immune responses and viral control post-ART only in animals categorized as ART responders, defined as sustaining median viral loads < 5 x 104 RNA copies/mL of plasma during ART. While the animals in this study also showed a similarly wide range of responses to ART, all were considered ART responders based on the criteria we used in our two previous therapeutic vaccine studies utilizing this virus. To address this concern, we added a supplemental figure with these animals excluded. The results shown in S9 Fig show that excluding these animals does not impact our conclusions, likely due to the small group sizes. The following text to the manuscript (Results, Page 13, Lines 291-301):

In our previous therapeutic vaccine study, we excluded animals with high levels of persistent viral replication on ART from our analyses and observed a significant impact of therapeutic vaccination only in animals categorized as ART responders (Median viral loads < 5 x 104 RNA copies/mL of plasma during ART) [39]. Although the animals in this study demonstrated a wide range of virologic suppression on ART, all were considered ART responders based on the criteria used in our two previous therapeutic vaccine studies utilizing SIVΔB670. However, to address the possibility that the two animals with incomplete viral suppression on ART could be skewing our results, we added a supplemental figure with A16234 from the mock-vaccinated group and A16236 from the MAG + AC group excluded (S9 Fig). This demonstrates that excluding these animals from analysis of vaccine efficacy did not impact our conclusions, likely due to the small group sizes.

2. Along those lines, the IFN-γ ELISpot data showed a significant difference in responses between MAG + AC and MAG + LT vaccine groups at 50 wpi (Fig 2B-C). However, the ICS data showed no difference between vaccine groups at any time point (S2-5 Figs). The reason for this discrepancy is unclear and was not discussed? Also, why were SIV Pol responses not readily detectable in either vaccine group considering Pol was included in the vaccine construct?

Response: The discrepancy between the ELISpot and ICS data is due to the fact that the ELISpot data represents the IFN-γ response in PBMCs as a whole, whereas the ICS data analyzes the CD4+ and CD8+ T cell responses separately. If the CD4+ and CD8+ IFN-γ T cell responses are combined, the trends in increased IFN-γ responses measured by ICS and ELISpot are consistent but not identical. This is likely due to differences in how the responses are measured with these two assays. ICS measures the percent of the CD4+ and CD8+ T cell population that are producing IFN-γ, whereas ELISpot reports the fraction of cells producing IFN-γ out of all PBMCs, not just T cells. In summary, when CD4+ and CD8+ T cell responses that were analyzed separately via ICS are combined, the overall IFN-γ T cell responses in each group are consistent with the ELISpot results. However, the relative differences between groups are not identical likely due to significant differences in the populations of cells analyzed by each assay, the assay protocols, and their readouts. We have added a supplementary figure and the following lines to the text to clarify the differences between these assays. (Results, Page 12, Lines 258-265):

In contrast to the IFN-γ ELISpot assay, where we analyzed the SIV-specific IFN-γ response as a whole in bulk PBMCs, here we assessed CD4+ and CD8+ T cell effector functions separately. There were no significant differences between groups in these functions in either the PBMC or MLN (S2-S5 Figs), although the MAG + AC group demonstrated a trend towards higher cumulative IFN-γ+ CD4+ and CD8+ T cells in the PBMC (S6 Fig). This trend was not statistically significant, likely due to the significant differences in the populations of cells analyzed by each assay. However, the ICS analysis affirms our ELISpot results are representative of the overall IFN-γ+ T cell response. 

Response: Responses to Pol were detected in 2/5 animals in both the MAG + LT and MAG + AC groups. These results were relatively low in comparison to Gag-specific and Env-specific responses, so they were originally grouped in with the Accessory Protein-specific responses. To resolve this ambiguity, we edited the figure to display Pol-specific responses separately and adjusted the text of the results section accordingly. (Results, Page 11, Lines 248-254):

Responses to Pol were detected in two out of five animals in both the MAG + LT and MAG + AC groups, accounting for up to 20% of the total IFN-γ response (Fig 2F). Env-specific responses were detected in two out of five animals in the MAG + LT group and one out of five animals in the MAG + AC group, with up to 15.5% of the total IFN-γ response targeting Env. In contrast, IFN-γ responses in the mock-vaccinated group were only detected in two out of four animals and predominantly targeted either Pol or accessory proteins (Vif, Rev and Nef), with Gag-specific responses detected in only one animal (Fig 2F).

3. The comparison of CD8+ T cell polyfunctional responses was unclear. What proportion of CD8+ T cell responses in controllers were not polyfunctional? Also, did the vaccine improve polyfunctionality of the CD8+ T cell response? A treatment group comparison of CD8+ T cell polyfunctional responses should have been presented to bolster the correlational analysis and resultant conclusions.

Response: We added a supplemental figure comparing the proportion of polyfunctional responses to non-polyfunctional responses. The data shows that there are no differences between controllers and non-controllers in non-polyfunctional CD8+ T cell responses, although the majority of the SIV-specific CD8+ T cell response is not polyfunctional. This suggests that polyfunctional CD8+ T cells play a greater role in controlling viral replication than non-polyfunctional CD8+ T cell responses. The following lines were added to the manuscript to address these points (Results, Page 17, Lines 380-387):

Notably, there are no differences between controllers and non-controllers in the frequencies of CD8+ T cells expressing one or two effector functions (non-polyfunctional responses) before or during ATI (S13 Fig). Although non-polyfunctional responses make up the majority of the SIV-specific CD8+ T cell response, these results demonstrate that polyfunctional CD8+ T cells likely play a greater role in controlling viral replication. 

Response: We also added a supplemental figure where we compare the frequencies of polyfunctional responses pre- and post-vaccination in the treatment groups. These results show 3/5 of the MAG + AC animals exhibited increased SIV-specific polyfunctional CD8 T cell responses after vaccination when compared to their pre-vaccination levels, but these results were not statistically significant. Additional clarification regarding these comparisons has been added to the text as follows. (Results, Page 12, Lines 266-273):

To further characterize the SIV-specific CD8+ T cell response post-vaccination, we assessed the magnitude of the polyfunctional CD8+ T cell response. Here, we define polyfunctionality as the frequency of T cells specific for either Gag or Env and expressing any three or more of the cytokines IFN-γ, TNFα, IL-2, and/or co-expressing the cytolytic markers CD107a/Granzyme B. Three out of five animals in the MAG + AC group exhibited increases in SIV-specific polyfunctional CD8+ T cells post-vaccination in both PBMC and MLN, whereas the frequency of polyfunctional CD8+ T cells in both the MAG + LT and mock-vaccinated controls remained nearly undetectable. However, these results were not statistically significant (S7 Fig). 

4. Again, it is difficult to make the case that mucosal immune responses prior to ART suppression influence virologic outcome during and post-ART (Fig 7) if the immune corelate analysis includes animals with ongoing viremia during ART. Is it unclear how the frequencies of colonic CD4+ T cells or lower Th17/Treg ratios prior to infection would affect the ability of ART to suppress viremia. By including the 2 animals with incomplete virus suppression, it is difficult to reach any of the conclusions outlined. 

Response: We thank the reviewer for pointing out the need for clarification in our discussion of Fig 7 but respectfully disagree that inclusion of the two animals with ongoing viremia during ART detracts from our analysis. As our analysis investigated the inherent associations between colonic CD4+ T cells and Th17/Treg ratios prior to infection and the virologic outcome during ATI, it is important to include these animals since both effective and ineffective viral control during ART must be considered in evaluating inherent factors influencing this variable outcome. We modified this section of the results to better explain our rationale for this analysis and conclusions (Results, Page 18, Lines 402-406):

Importantly, lower acute viral burden was associated with lower residual viral replication during ART (P = 0.0078, Fig 7B), and lower viral burden on ART was strongly correlated with better control of viral replication during ATI (P = 0.00040, Fig 7C). Together, these data suggest that pre-infection host immune parameters may have influenced the extent of acute viral replication and subsequently affected the extent of residual viral replication on ART, or ART efficacy. 

5. Another major issue is the fact that the controllers had lower viral loads at the time of ART initiation (Fig 4). This would suggest that these animals already had effective T cell responses prior to ART initiation, which were maintained during therapy and helped facilitate better post-ART control. So, the suggestion that “pre-infection immune parameters helped animals develop polyfunctional CD8+ T cell responses during ART” (lines 339 – 342) is not supported by the data presented.

Response: We appreciate the reviewer raising this point and recognize the importance of not overstating our conclusions. As shown in the supplemental figure we added in response to comment 3, prior to vaccination, polyfunctional CD8+ T cell responses were not detectable in the PBMC of controllers, although they were detectable at low levels in the MLN of 4/5 controllers and 2/4 non-controllers. In controllers, the frequency of polyfunctional responses in both the PBMC and MLN increased after vaccination, though these results were not statistically significant. We have revised the results and discussion to be more precise (Results, Page 18, Lines 407-409):

Persistent viral replication on ART in turn may have influenced the ability of each animal to develop and maintain polyfunctional CD8+ T cell responses in the blood and GALT that significantly correlated with control of viral replication during ATI (Fig 6A-D).

(Discussion, Page 23 Lines 524-527):

Furthermore, the disruption of mucosal Th17 and Treg homeostasis, coupled with persistent, low-level viremia during ART in the non-controllers likely compromised their ability to develop or maintain polyfunctional CD8+ T cell responses in the blood and GALT and their subsequent failure to control virus during ATI. 

Minor concerns

The manuscript should be checked for grammatical errors. 

The manuscript has been proofread again and grammatical errors have been addressed.

Line 96 needs clarification as is says PD-1 and CTLA4 can “restore” immune exhaustion.

“Restore” has been edited to “reverse.”

Line 268 should be “S9 Fig.” not “S7 Fig.”

This has been corrected.

Line 277 says “S6 Fig.” but the figure which is supposed to show the correlation between viral control and Gag-specific CD4+ T cell responses is missing.

This has been corrected.

There was no data or discussion on the impact of therapeutic vaccination on cell-associated viral loads. Did the controllers have a lower reservoir size? If so, perhaps this could also contribute to improved post-ART viral control.

Response: We thank the reviewer for raising this point and have added the following paragraph to the discussion to address the possible effects of the latent reservoir on post-ART viral control. (Discussion, Pages 22-23 Lines 502-508):

Another factor that could have influenced post-ART viral rebound is the size of the latent reservoir. Previous studies showed levels of proviral DNA in the GALT correlate with time to viral rebound [62], and increased proviral DNA in PBMC correlates with higher viral loads during ATI [63]. It is therefore possible that the controllers in this study had reduced proviral reservoirs that contributed to improved control of virus replication. Further experiments are needed to quantify cell-associated viral DNA in the GALT and PBMC and to determine how the latent reservoir may have affected virologic outcome during ATI.

---

## [Decision Letter · Decision Letter 1]

6 May 2021

PONE-D-21-01499R1

Effects of therapeutic vaccination on the control of SIV in rhesus macaques with variable responsiveness to antiretroviral drugs.

PLOS ONE

Dear Dr. Fuller,

Thank you for submitting your manuscript to PLOS ONE. After careful consideration, we feel that it has merit but does not fully meet PLOS ONE’s publication criteria as it currently stands. Therefore, we invite you to submit a revised version of the manuscript that addresses the points raised during the review process.

Please see below; both reviewers raised some minor points; once addressed, this manuscript can be accepted for publication.

We look forward to receiving your revised manuscript.

Kind regards,

Siddappa N. Byrareddy, PhD

Academic Editor

PLOS ONE

Journal Requirements:

Reviewers' comments:

Reviewer's Responses to Questions

**Comments to the Author**

1. If the authors have adequately addressed your comments raised in a previous round of review and you feel that this manuscript is now acceptable for publication, you may indicate that here to bypass the “Comments to the Author” section, enter your conflict of interest statement in the “Confidential to Editor” section, and submit your "Accept" recommendation.

Reviewer #1: (No Response)

Reviewer #2: All comments have been addressed

2. Is the manuscript technically sound, and do the data support the conclusions?

Reviewer #1: Yes

Reviewer #2: Partly

3. Has the statistical analysis been performed appropriately and rigorously? 

Reviewer #1: Yes

Reviewer #2: I Don't Know

4. Have the authors made all data underlying the findings in their manuscript fully available?

Reviewer #1: Yes

Reviewer #2: Yes

5. Is the manuscript presented in an intelligible fashion and written in standard English?

Reviewer #1: Yes

Reviewer #2: Yes

6. Review Comments to the Author

Reviewer #1: This revised manuscript compares the therapeutic impact of DNA-based vaccines encoding SIV Gag, Pol, and Env antigens mixed with either a single adjuvant or combination of adjuvants in ART-treated SIV-infected rhesus macaques. The vaccines had no impact on viral rebound post-ART, but more MAG-AC immunized macaques controlled virus replication below 1x10^3 SIV copies/ml than MAG-LT or unimmunized controls. Immune parameter analysis of controllers in all groups suggests that polyfunctional SIV-specific CD8 T cells contributed to virus suppression. Although, this analysis is complicated by the partial resistance of SIV∆670 to ART, potentially preserving polyfunctional T cell responses in ART responders versus non-responders. Further, this study reports pre-infection mucosal CD4 T cell levels and Th17/Treg balances may have long-term impacts on disease progression and HIV cure strategies.

1) In figure 2F, all IFN-g responses at week 50 in A16236 are against accessory proteins, indicating that MAG+AC immunization did not amplify Gag, Pol, or Env responses. These results should be briefly discussed in the text.

2) Line 484, remove “strong” from this sentence. The data suggest that polyfunctional CD8 T cell responses contribute to controlling virus post-ART but labeling the evidence as “strong” is overstating the data.

Reviewer #2: While the authors have addressed most of my concerns, to fully define how intrinsic resistance factors did not play a role in virology outcome, a table showing animal demographics including MHC typing, TRIM5 genotype and viremia status (controller or non-controller) should be included.

7. PLOS authors have the option to publish the peer review history of their article (what does this mean?). If published, this will include your full peer review and any attached files.

Reviewer #1: No

Reviewer #2: No

---

## [Author Response · Author response to Decision Letter 1]

9 May 2021

We thank the reviewers for their continued constructive critique and appreciate the additional opportunity to refine this manuscript. To address each concern, we edited the text as indicated below and added new supplemental data where needed. We believe these revisions fully address the reviewers’ concerns and further improve the focus and impact of the manuscript.

Reviewer 1 Comments:

1) In figure 2F, all IFN-g responses at week 50 in A16236 are against accessory proteins, indicating that MAG+AC immunization did not amplify Gag, Pol, or Env responses. These results should be briefly discussed in the text.

We agree that this is an important detail that should be addressed. The following text was added (Results, Page 11, Lines 253-256):

SIV-specific IFN-γ responses were not observed in one animal in the MAG + LT group (A16150), while one animal in the MAG + AC group (A16236) only exhibited IFN-γ responses to accessory proteins, indicating vaccination had no effect on the SIV-specific IFN-γ response in 1/5 animals in each vaccine group. 

2) Line 484, remove “strong” from this sentence. The data suggest that polyfunctional CD8 T cell responses contribute to controlling virus post-ART but labeling the evidence as “strong” is overstating the data.

We recognize the importance of presenting our data objectively and removed the word “strong” from the indicated sentence (Page 22, Line 488 after revisions).

Reviewer 2 Comments: 

1) While the authors have addressed most of my concerns, to fully define how intrinsic resistance factors did not play a role in virology outcome, a table showing animal demographics including MHC typing, TRIM5 genotype and viremia status (controller or non-controller) should be included.

We recognize that certain TRIM5 haplotypes and MHC-1 alleles are associated with improved control of viral replication and can be a confounding factor in vaccine studies. To address this, we included a supplemental table showing each animal’s TRIM5 genotype, virological status, vaccine group, and indicating the presence or absence of MHC-1 alleles associated with lower viral loads. The following text was also edited (Results, Page 14, Lines 317-319): 

Viral burden during ATI was significantly different between controllers and non-controllers (P = 0.0010, Fig 4B). Certain MHC and TRIM5 genetics have been associated with improved viral control in SIV infected macaques [52-57], however, we observed no association between these MHC or TRIM5 alleles and the control of viral rebound (S1 Table) indicating these genotypes did not likely influence viral burden during ATI in this study.

---

## [Editor Report · Decision Letter 2]

2 Jun 2021

Effects of therapeutic vaccination on the control of SIV in rhesus macaques with variable responsiveness to antiretroviral drugs.

PONE-D-21-01499R2

Dear Dr. Fuller,

We’re pleased to inform you that your manuscript has been judged scientifically suitable for publication and will be formally accepted for publication once it meets all outstanding technical requirements.

Kind regards,

Siddappa N. Byrareddy, PhD

Academic Editor

PLOS ONE
---

## [Editor Report · Acceptance letter]

7 Jun 2021

PONE-D-21-01499R2 

Effects of therapeutic vaccination on the control of SIV in rhesus macaques with variable responsiveness to antiretroviral drugs 

Dear Dr. Fuller:

I'm pleased to inform you that your manuscript has been deemed suitable for publication in PLOS ONE. Congratulations! Your manuscript is now with our production department. 

Kind regards, 

on behalf of

Dr. Siddappa N. Byrareddy 

Academic Editor

PLOS ONE